# CaBLAM: a high-contrast bioluminescent Ca²⁺ indicator derived from an engineered *Oplophorus gracilirostris* luciferase

Gerard G. Lambert[1,2,17], Emmanuel L. Crespo[3,17], Jeremy Murphy [4,17], Kevin L. Turner [4], Emily Gershowitz[5,6], Michaela Cunningham [4], Daniela Boassa [1], Selena Luong[7], Dmitrijs Celinskis[4], Justine J. Allen [4], Stephanie Venn[3], Yunlu Zhu[5,6], Mürsel Karadas[5,8], Jiakun Chen[9], Roberta Marisca[10], Hannah Gelnaw [5,6], Daniel K. Nguyen[7], Junru Hu[11], Brittany N. Sprecher [1], Maya O. Tree[3], Richard Orcutt[1], Daniel Heydari[1], Aidan B. Bell[7], Albertina Torreblanca-Zanca [1], Ali Hakimi[12], Tim Czopka[10,13], Shy Shoham[5,8,14,15], Katherine I. Nagel[5,14], David Schoppik [5,6,14], Arturo Andrade[4], Diane Lipscombe[4], Christopher I. Moore [4], Ute Hochgeschwender[3] & Nathan C. Shaner [1,2,16] ✉

Monitoring intracellular calcium is central to understanding cell signaling across nearly all cell types and organisms. Fluorescent genetically encoded calcium indicators (GECIs) remain the standard tools for in vivo calcium imaging, but require intense excitation light, leading to photobleaching, background autofluorescence and phototoxicity. Bioluminescent GECIs, which generate light enzymatically, eliminate these artifacts but have been constrained by low dynamic range and suboptimal calcium affinities. Here we show that CaBLAM ('calcium bioluminescence activity monitor'), an engineered bioluminescent calcium indicator, achieves an order-of-magnitude improvement in signal contrast and a tunable affinity matched to physiological cytosolic calcium. CaBLAM enables single-cell and subcellular activity imaging at video frame rates in cultured neurons and sustained imaging over hours in awake, behaving animals. These capabilities establish CaBLAM as a robust and general alternative to fluorescent GECIs, extending calcium imaging to regimes where excitation light is undesirable or infeasible.

Genetically encoded Ca²⁺ indicators (GECIs) play a central role in biomedical research, with much of our current understanding of systems neuroscience based on Ca²⁺ imaging. GECIs based on fluorescent proteins have been under continuous development for more than two decades, yielding diverse lineages with progressively improved brightness, kinetics and dynamic range[1–8]. Among these, the GCaMP[8] and GECO[9] families represent closely related and heavily optimized designs that have influenced most modern Ca²⁺ indicators. The GCaMP family, now

in its eighth generation, exemplifies this iterative progress and remains a benchmark for fluorescent protein-based GECI performance.

While powerful, fluorescence imaging has key limitations. Most importantly, obtaining fluorescent signals requires high intensity photon excitation. This bombardment typically causes photobleaching and high background autofluorescence, undermining sensitivity and spatial precision. Intense excitation can also cause photodamage, severely limiting the long-term (for example, whole lifetime) imaging

SSLuc mutations

CaBLAM model

| SSLuc (1–156) | RS20 | 5 × GGS | CaM (N98I) | VTGYRLLEEISN |
|---|---|---|---|---|

SASS SPA

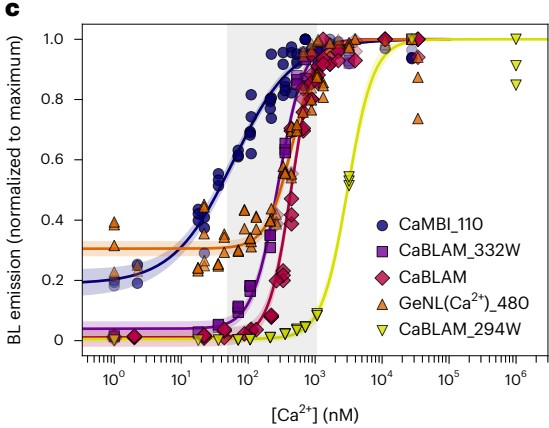

**Fig. 1 | CaBLAM architecture and Ca²⁺ affinity. a**, Structure models of eKL9h and CaBLAM. Side chain positions with mutations relative to eKAZ are depicted as yellow sticks in the luciferase model. The C-terminal peptide is depicted in magenta on both structure models. **b**, Architecture of the sensor component of CaBLAM (not to scale), with colors corresponding to the model structure backbone. Linkers between the split luciferase components and the sensor domain are labeled with their one-letter amino acid sequences, SASS (Ser-Ala-Ser-Ser) and SPA (Ser-Pro-Ala). Full-length CaBLAM includes the N-terminal mNeonGreen FRET acceptor. **c**, In vitro Ca²⁺ titration of CaBLAM ($n = 6$, diamonds) alongside GeNL(Ca²⁺)_480 ($n = 8$, triangles), CaMBI_110

($n = 6$, circles) and two CaBLAM variants, 294W ($n = 3$, inverted triangles) and 332W ($n = 3$, squares), with altered Ca²⁺ affinities. The given $n$ value in parentheses for each BL GECI represents the number of technical replicates of the full titration performed. All individual data points are shown. BL intensities were normalized to the maximum BL within each replicate dataset, pooled for each indicator. Fit curves represent the mean global fit to a three-parameter Hill model, with shaded regions indicating the 95% confidence interval of the fitted mean. The typical physiologically relevant cytosolic Ca²⁺ concentration range between ~50 nM and ~1 μM is shaded in gray.

of cells. The optics required to deliver this light add hardware and fix the light path, challenges that are especially troublesome in experiments with freely moving animals. Bioluminescent (BL) GECIs remove these hurdles by generating light through an enzyme-driven reaction. The approach began with the natural *Aequorea victoria* photoprotein, aequorin[10–12], and has since grown to include a number of luciferase-based BL GECIs[13–18]. Despite these advances, no available BL GECI has achieved imaging performance comparable to highly optimized fluorescent indicators such as the GCaMP series. Existing BL GECIs typically afford limited dynamic range in vivo owing to high baseline emission and Ca²⁺ affinities that reduce sensitivity in the physiological range, restricting their practical application to population-level recordings.

Here, we describe the development of SSLuc (sensor scaffold luciferase), a variant of *Oplophorus gracilirostris*[19] luciferase (OLuc) engineered to provide increased BL emission in vitro along with improved folding and performance in protein fusions. SSLuc is several-fold more active than NanoLuc in vitro using the widely employed furimazine (Fz) substrate and is less prone to aggregation in cells than NanoLuc when fused to other proteins. We found that SSLuc was highly amenable to sensor domain insertion, making it a favorable scaffold for engineering a BL GECI, the calcium bioluminescence activity monitor (CaBLAM). Optimization of the C-terminal peptide sequence was a critical component of both luciferase and GECI engineering, and targeted mutations to the calmodulin (CaM) domain of the sensor were particularly important for obtaining high contrast in CaBLAM.

CaBLAM has similar maximum light output to other described BL GECIs in cells but displays ~83-fold full-range contrast in vitro and up to 15–20-fold in live cultured cells in response to physiological changes in cytosolic Ca²⁺ levels. We further demonstrate that CaBLAM reliably reports the Ca²⁺ signal from a field stimulation equivalent to a single action potential when imaged at single-cell resolution at 10 Hz in cultured primary neurons on a typical widefield microscope and EMCCD camera. In mice, CaBLAM was also readily imaged at 10 Hz in head-fixed awake animals, where its ability to report physiologically relevant neural Ca²⁺ activity evoked by vibrissa stimulation at the single-cell level was superior to GCaMP6s under widefield single-photon (1-photon) illumination. In zebrafish, we demonstrate CaBLAM detection in several

genetically targeted transgenic lines in awake head-fixed larvae, with sampling rates of 40 Hz using a photomultiplier tube (PMT) or 20 Hz using an intensified camera. In both cases, CaBLAM produced robust signals following large tail movements with kinetics tuned to cell type.

## Results

### Directed evolution of high-activity soluble OLuc variants
We initially set out to develop a variant of OLuc that could more efficiently use coelenterazine (CTZ), the native substrate of many marine luciferases including OLuc, reasoning that NanoLuc was only one of the possible endpoints achievable through directed evolution of this lineage of luciferases. Starting from the OLuc mutant eKAZ[20], we performed multiple rounds of directed evolution consisting of alternating error-prone and site-directed mutagenesis libraries in *Escherichia coli*. For each round, variants were selected based on higher BL emission and solubility. Subsequent engineering of the intact luciferase included the generation of mNeonGreen[21] fusions for enhancement of emission quantum yield, incorporation of a subset of mutations found in NanoLuc[22] and NanoBit[23], optimization of the C-terminal peptide sequence, and fine-tuning interactions between the C-terminal peptide and the rest of the protein. The final clone, which we named SSLuc (Fig. 1a), can be extracted with nearly 100% efficiency from *E. coli* cultures, retains activity in vitro over longer storage periods than NanoLuc and is monomeric in mammalian cells. Contrary to our original intentions, SSLuc performs best using Fz as its luciferin, although its activity with other CTZ analogs is also higher than we measure for NanoLuc. We found that SSLuc does not appear brighter than NanoLuc when expressed in mammalian cells, despite its high performance in vitro, suggesting that availability of the luciferin substrate (for example, Fz) may limit SSLuc's maximum brightness in cells. SSLuc proved to be a favorable luciferase scaffold from which to generate biosensors, and led us to develop a high-performance BL GECI, as described below. A detailed account of our development and characterization of SSLuc can be found in Supplementary Text, Supplementary Table 1 and Supplementary Figs. 1–4.

### Development and evaluation of a high-contrast BL GECI
To engineer a high-contrast BL GECI, we split SSLuc in the loop joining its final two β-strands and inserted Ca²⁺-sensing domains, testing

**Table 1 | Properties of selected intensiometric genetically encoded Ca²⁺ indicators**

| GECI name | Type | $K_D$ (nM)[a] | Hill coefficient[b] | Contrast[c] | Ref. |
|---|---|---|---|---|---|
| GCaMP8s | FP | 46 | 2.2 | 50 | 8 |
| GCaMP8f | FP | 334 | 2.1 | 79 | 8 |
| NEMOf | FP | 528 | 3.2 | 246 | 4 |
| NEMOc | FP | 557 | 3.8 | 422 | 4 |
| CaMBI_110 | Luciferase | 59±6 (110) | 0.97±0.06 (2.5) | 4.8±0.1 (8) | 18 |
| GeNL(Ca²⁺)_480 | Luciferase | 457±20 (480) | 2.28±0.21 (3) | 3.6±0.3 (5) | 17 |
| CaBLAM | Luciferase | 439±14 | 2.62±0.18 | 83±9 | This work |
| CaBLAM_294W | Luciferase | 3067±95 | 2.24±0.19 | 624±39 | This work |
| CaBLAM_332W | Luciferase | 281±9 | 2.26±0.14 | 68±5 | This work |

[a]Binding constant for Ca²⁺ for FP-based GECIs as reported in the original publications; binding constant for Ca²⁺ as measured in vitro for CaMBI, GeNL(Ca²⁺)_480, CaBLAM and CaBLAM variants and (in parentheses) as reported in the original publications. [b]Hill coefficient (cooperativity) for FP-based GECIs as reported in the original publications; Hill coefficient for luciferase-based GECIs calculated from sigmoidal curve fit to Ca²⁺ titration series and (in parentheses) as reported in the original publications. [c]Contrast between Ca²⁺-free and Ca²⁺-saturated indicator in vitro for FP-based GECIs as reported in the original publications; contrast for luciferase-based GECIs measured in this work and (in parentheses) as reported in the original publications. Error margins represent 95% confidence interval (CI) for $K_D$ and Hill coefficient and s.e.m. for contrast, as measured in this work.

circular permutations and direct fusions of RS20–CaM, CaM–RS20 and Troponin C (Supplementary Fig. 5). All prototypes responded robustly in vitro, but the RS20–CaM topology (Fig. 1a,b) produced the largest BL change (Fig. 1a,b), echoing a hybrid NanoBit-GCaMP-based design (GLICO[14]) while retaining SSLuc's fixed N-terminal fluorescent partner for Förster resonance energy transfer (FRET) enhancement[17] and restricting Ca²⁺ readout to the BL channel. Early clones had very high Ca²⁺ affinity (dissociation constant $K_D$ < 10 nM), so we iteratively tuned affinity and contrast by swapping RS20/CaM sequences from fluorescent GECIs[24–26], shortening or extending inter-domain linkers, and introducing rational EF-hand mutations (Supplementary Fig. 5). The optimal combination we identified used the GCaMP6s RS20/CaM pair combined with a naturally occurring CaM mutation, N98I[27], that lowers the apparent Ca²⁺ affinity (Fig. 1c and Table 1), aligning the sensor's most sensitive range with typical cytosolic Ca²⁺ levels observed in hippocampal neurons[28]. To maximize contrast, we next systematically evaluated variants in this framework using the collection of C-terminal peptides we had generated while developing SSLuc (Supplementary Text and Supplementary Table 1), finding that the low-affinity sequence VTGYRLFEEIL ('pep114', ref. 23) produced the greatest increase in BL emission between low and high Ca²⁺ concentrations. Based on these results, we redesigned the optimized C-terminal peptide from SSLuc to arrive at VTGYRLLEEISN, which preserves the catalytic Arg while replacing Lys residues with Glu to weaken peptide–enzyme binding and retain high contrast. This final architecture (Fig. 1b and Supplementary Fig. 5d) was dubbed CaBLAM.

We characterized purified CaBLAM protein in vitro, determining a $K_D$ of ~439 nM for Ca²⁺ binding and an absolute contrast (that is, the ratio of luminescence emission between saturating Ca²⁺ and zero Ca²⁺, typically reported for this class of sensor) of ~83-fold (Fig. 1c and Table 1). This value places CaBLAM favorably relative to other BL GECIs published to date, all of which display substantially lower contrasts than CaBLAM[14,17,18]. The Hill coefficient of CaBLAM is ~2.6, similar to that of the luminescent reporter for calcium signaling GeNL(Ca²⁺)_480 (ref. 17) (reported as 3 (ref. 17), measured at ~2.3 in this study) and notably higher than that of CaMBI[18], which displays a Hill coefficient of ~1 in our hands (Fig. 1c and Table 1). We also characterized two additional EF-hand mutants of CaBLAM that retain high brightness and contrast but have higher and lower Ca²⁺ affinities (Table 1 and Supplementary Data 1): 294W (N98W in CaM; $K_D$ ~3 μM) and 332 W (Q135W in CaM; $K_D$ ~280 nM).

Since CaBLAM has a similar Ca²⁺ affinity to GeNL(Ca²⁺)_480 and shares the same mNeonGreen FRET acceptor, we chose these two BL GECIs as a matched pair for benchmarking in cells. In practice, when a GECI is expressed in a living system, its contrast typically drops substantially relative to in vitro values, and so we next evaluated the behavior of CaBLAM in cultured cells in comparison with GeNL(Ca²⁺)_480 to determine how much of its promising in vitro behavior would translate to improved real-world performance.

**Characterization and benchmarking of CaBLAM in cells**

CaBLAM markedly outperformed the NanoLuc-derived indicator GeNL(Ca²⁺)_480 across cell lines and primary neurons in terms of contrast. In U2OS cells treated with ionomycin, CaBLAM produced ~5.4-fold higher contrast between resting and high Ca²⁺ versus GeNL(Ca²⁺)_480 (Supplementary Fig. 6a), owing to CaBLAM's low baseline luminescence and favorable Hill coefficient. CaBLAM's performance in HeLa cells showed similar advantages, generating ʟ-histamine-induced BL oscillations that were ~3.3-fold larger than those for GeNL(Ca²⁺)_480 and displaying diverse oscillatory phenotypes that ranged from seconds to minutes (Supplementary Fig. 6b,d and Supplementary Video 1). In HeLa cells, CaBLAM exhibited approximately 40% of the per-molecule BL brightness of GeNL(Ca²⁺)_480 (Supplementary Text and Supplementary Fig. 6), with high cell-to-cell variability likely arising from rate-limiting substrate diffusion across the plasma membrane. Primary rat cortical neurons exhibited ~20-fold bioluminescence increase after KCl depolarization versus ~4-fold with GeNL(Ca²⁺)_480 (Supplementary Fig. 6c), leading us to begin exploring CaBLAM's performance relative to fluorescent GECIs as well.

**Comparison of CaBLAM to GCaMP8s in rat hippocampal neurons**

We performed simultaneous Ca²⁺ imaging and electrical field stimulation to characterize the CaBLAM sensor (Supplementary Video 2) and to benchmark it against the state-of-the-art fluorescent sensor GCaMP8s in rat hippocampal neurons. CaBLAM-expressing neurons showed low detectable photon counts during 10 Hz imaging, yet individual traces demonstrated detectable evoked responses across a range of electrical field stimulations (Fig. 2a,c). As expected for a fluorescent indicator, GCaMP8s was brighter overall and captured both spontaneous and evoked neural activity (Fig. 2b,d). These initial observations represent typical examples of the responses observed for each indicator. We calculated the evoked change in bioluminescence for CaBLAM, showing that it reliably reported changes in intracellular Ca²⁺, with large changes observed in the normalized ($\Delta L/L$) signal (Fig. 2c). For GCaMP8s (Fig. 2d), the normalized change in fluorescence ($\Delta F/F$) was reduced due to its high baseline fluorescence. Despite this, GCaMP8s still produced observable changes in fluorescence, consistent with previous reports.

We next characterized stimulus-evoked bioluminescence and fluorescence signals in neurons expressing CaBLAM (Fig. 3a) and GCaMP8s

(Fig. 3b). Consistent with our initial observation, electrical field stimulations (1, 5 and 40 pulses at ~83 Hz, 1 ms pulse) generated higher detectable changes in $\Delta L/L$ with CaBLAM when compared to GCaMP8s $\Delta F/F$ due to the low baseline signal of CaBLAM (Fig. 3c, Wilcoxon signed rank sum test, two-tailed, Bonferroni correction for multiple comparisons, CaBLAM versus GCaMP8s at 1 field stimulation: $P = 1.1 \times 10^{-10}$, 5 field stimulations: $P = 2.45 \times 10^{-21}$, 60 field stimulations: $P = 2.03 \times 10^{-2}$). The response kinetics of CaBLAM are considerably slower than GCaMP8s, as expected, since CaBLAM has not yet been optimized for fast responses (Fig. 3d, two-tailed Kolmogorov–Smirnov test, $P = 7.1 \times 10^{-51}$, $n = 96$ trials for 20–38 GCaMP8s-expressing neurons versus 570 trials for 88–276 CaBLAM-expressing neurons).

For both indicators, we measured the time to peak $\Delta L/L$ (CaBLAM) and $\Delta F/F$ (GCaMP8s) and observed distinct response timing. GCaMP8s reached peak responses around 100 ms (approximately 1 frame post-stimulation at our 10 Hz imaging rate) for 1 and 5 pulses, while the response extended to 300 ms (about three frames) for higher stimulation frequencies (Fig. 3e, Wilcoxon signed rank sum test, two-tailed, Bonferroni correction for multiple comparisons, CaBLAM versus GCaMP8s at 1 field stimulation: $P = 1.93 \times 10^{-5}$, 5 field stimulations: $P = 1.07 \times 10^{-16}$, 60 field stimulations: $P = 1.24 \times 10^{-22}$). In contrast, CaBLAM under constant Fz perfusion (9.2 μM final concentration), displayed median peak response times of approximately 700 ms, 800 ms and 1500 ms poststimulation for 1, 5 and 40 pulses, respectively. As these experiments did not include synaptic blockers, the true indicator kinetics are likely faster than measured. Next, we compared the peak signal to noise ratio (SNR) between CaBLAM and GCaMP8s. As we expected when imaging a monolayer of primary neurons in culture, GCaMP8s exhibited higher peak SNR in this scenario despite its lower contrast relative to CaBLAM (Fig. 3f, Wilcoxon signed rank sum test, two-tailed, Bonferroni correction for multiple comparisons, CaBLAM versus GCaMP8s at 1 field stimulations: $P = 7.86 \times 10^{-11}$, 5 field stimulations: $P = 1.14 \times 10^{-19}$, 40 field stimulations: $P = 1.41 \times 10^{-14}$). CaBLAM exhibited larger and faster stimulus-evoked BL responses at 4.6 μM than at 9.2 μM Fz (Extended Data Fig. 1a–c), indicating improved sensitivity at the lower substrate concentration. However, higher substrate (9.2 μM) restored the proportion of responsive neurons to levels comparable to GCaMP8s (Extended Data Fig. 1d), supporting 9.2 μM as the optimal Fz concentration for reliable detection of evoked activity.

## Comparison of CaBLAM and GCaMP6s in vivo

After determining that fluorofurimazine (FFz) was the optimal substrate for CaBLAM in cultured cells (Supplementary Text and Supplementary Figs. 8 and 9), we next characterized and compared the performance of CaBLAM to GCaMP6s in vivo using an established model for sensor development[29], neocortical interneurons labeled in the neuron-derived neurotrophic factor (NDNF)-Cre mouse line (Fig. 4 and Supplementary Video 3). After selective viral expression and surgical preparation, we performed imaging either without external illumination (CaBLAM) or under 1-photon epifluorescent illumination (GCaMP6s). All mice were head-fixed and free to run on a wheel as tactile stimuli were delivered to their vibrissae (Fig. 4a). Imaging was performed through a cranial window and, in the CaBLAM-expressing group, FFz was infused through a canula implanted at the edge of the imaging window (Fig. 4b,c). This approach allowed us to directly assess single-cell 1-photon BL and fluorescent signals evoked by tactile stimulation. All animals exhibited detectable activity in single interneurons (total cell numbers: CaBLAM: median 42, interquartile range (IQR) 18; GCaMP6s: median 84, IQR = 74.25) (Fig. 4d). Replicating recent findings, subsets of NDNF cells exhibited either positive (Fig. 4e,g–i) or negative responses to tactile stimuli[30], and modulation by arousal state as indexed by their running speed[31] (Fig. 4j). Most responsive cells in both sensor groups were positively modulated by tactile stimuli (Fig. 4f, CaBLAM 97% and GCaMP6s 72% of all responsive cells), and as

such we focused our analyses on these cells (see Extended Data Fig. 2 for examples of negative responses).

The tactile evoked waveforms averaged across neurons and animals exhibited close correspondence between the two sensor groups and, although necessarily different measures, tactile evoked $\Delta F/F_0$ and $\Delta L/L_0$ were highly comparable in magnitude and overall temporal dynamics (Fig. 4i). We measured response onset latency by finding the half peak within the 0–2 s response window (Fig. 4h). With this metric, CaBLAM onset latencies occurred later than GCaMP6s onset latencies (Wilcoxon rank sum, $P = 3.3 \times 10^{-8}$, $z = 5.52$; CaBLAM: $n = 82$, median 1 s, IQR = 0.7, GCaMP6s: $n = 44$, median 0.45 s, IQR = 0.25). This difference in response latency agrees with our in vitro data.

To compare the ability of each sensor to detect a signal when present, single-trial SNR was computed across all cells as $\max(\frac{\Delta X}{X_0})/\sigma(X_0)$, normalizing the maximum change from baseline 0 to 2 s poststimulus (numerator) by the standard deviation (s.d.) of the baseline period (denominator, −3 to 0 s). Single-trial CaBLAM SNRs were significantly higher than those of GCaMP6s (Wilcoxon rank sum test, $P = 1.89 \times 10^{-5}$, $z = 4.28$; CaBLAM SNR: $n = 3127$, median 2.18, IQR = 3.22; GCaMP SNR: $n = 1,570$, median 1.83, IQR = 3.20). To determine the components of the SNR that differ between the sensors, we ran two subsequent analyses comparing the single-trial peak values and single-trial baseline s.d. We found that while signal peaks were typically larger in CaBLAM, this trend was not significant ($P = 0.079$, $z = 1.76$; CaBLAM median 0.096 $\Delta L/L_0$, IQR = 0.16; GCaMP6s median 0.088 $\Delta F/F_0$, IQR = 0.17). However, CaBLAM baseline was steadier, showing a significantly smaller s.d. ($P = 1.23 \times 10^{-3}$, $z = -3.26$; CaBLAM median 0.044 $\Delta L/L_0$, IQR = 0.049; GCaMP6s median 0.049 $\Delta F/F_0$, IQR = 0.048). Thus, the lower background of BL signals compared with epifluorescent signals accounted for the significantly higher single-trial SNR.

We next sought to test whether CaBLAM was effective in reporting sensory-driven dynamics with peripheral administration of luciferin, a common experimental design. We injected 200 μl of CFz retro-orbitally in mice expressing CaBLAM pan-neuronally ($n = 3$; CaBLAM expressed by injection of AAV2/9 under control of the hSyn promoter). Imaging a wide field of view (3.23 mm across) at 2 Hz, we saw robust BL responses to tactile stimulation with rapid onset times. Calculated as the half-peak latency of the average response, all mice exhibited onset latencies at time 0, indicating that onset occurred in the 0.5-s window between stimulus onset and the first poststimulus image. This rapid onset time precedes almost all local hemodynamic responses (for example, dilation), a concern that has been shown to contribute to sensory responses in other BL GECIs[32] (Fig. 4k,l). Trial-to-trial SNR was significantly greater than 0 (one-sample Wilcoxon signed rank test, $P = 2.64 \times 10^{-30}$), if typically less than the single-cell NDNF response (median 1.83, IQR = 2.66).

To better define the rapid onset of this response, we subsequently imaged a mouse under the same conditions at 10 Hz and found a highly comparable sensory response with regards to shape and magnitude (Fig. 4l). At 10 Hz the trial-to-trial SNR of the animal's sensory response was slightly higher than at 2 Hz (10 Hz SNR: median 2.91, IQR = 1.89 versus 2 Hz SNR: median 2.36, IQR = 3.06) but not significantly different (Wilcoxon rank sum test, $P = 0.52$, $z = 0.64$). We also tested the efficacy of this administration route and concentration in an NDNF-Cre mouse expressing CaBLAM. At a 0.5-Hz frame rate and under the same conditions used for direct neocortical infusions, BL signals were detectable from NDNF cells but proved too weak to capture Ca²⁺ fluctuations on a relevant time scale.

To test the viability of CaBLAM as a long-duration sensor, FFz was applied directly to the exposed cortex of a pan-neuronal CaBLAM mouse under anesthesia, and BL recorded at 10 Hz for >5 h. Imaging commenced ~2 min after FFz addition, at which time the BL signal was already apparent. To evaluate the overall time course of BL intensity, as context for understanding the SNR of the evoked response, mean emission values from a circular region of interest (ROI)

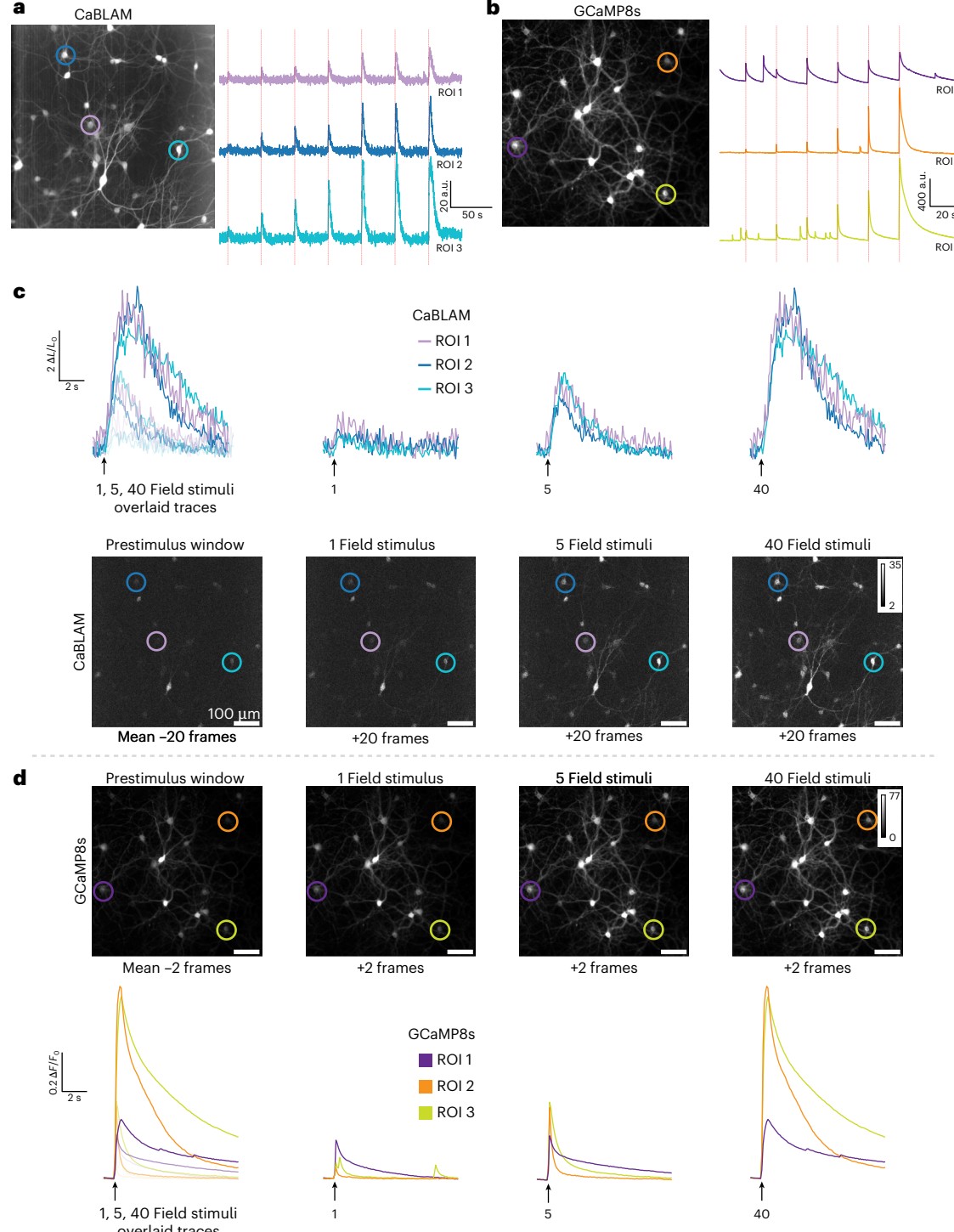

**Fig. 2 | Overview of a typical Ca²⁺ imaging session comparing electrical field evoked responses in CaBLAM- and GCaMP8-expressing rat hippocampal neurons. a,** Representative CaBLAM bioluminescence imaging session. Left: mean *z*-stacked image with ROIs highlighted by colored circles (ROI 1 in light purple, ROI 2 in blue, and ROI 3 in teal). Right: BL Ca²⁺ transients from ROIs 1–3 displaying responses evoked by multiple field stimuli (red dashed lines indicate field stimulation events). Similar responses were observed in at least four imaging sessions. **b,** Representative GCaMP8s fluorescence imaging session. Left: mean *z*-stacked image with individual ROIs highlighted by colored circles (ROI 1 in dark purple, ROI 2 in orange, ROI 3 in yellow). Right: fluorescence Ca²⁺ transients from ROIs 1–3 displaying responses evoked by multiple field stimuli (red dashed lines indicate stimulation events). For **a** and **b**, values are in arbitrary units (a.u.). Similar responses were observed in at least 3 imaging sessions. **c,** CaBLAM Ca²⁺ responses to varied field stimuli. Top: individual BL $\Delta L/L$ Ca²⁺ response traces from ROIs 1–3 for each field stimulus condition. Bottom: mean *z*-stacked frames (20 frames per condition) for prestimulus, 1 field stimulus, 5 field stimuli and 40 field stimuli conditions, corresponding to data shown in the top panel. **d,** GCaMP8s Ca²⁺ responses to varied field stimuli. Top: individual fluorescence $\Delta F/F$ Ca²⁺ response traces from ROIs 1–3 for each stimulus condition. Bottom: mean *z*-stacked frames (2 frames per condition) for prestimulus, 1 field stimulus, 5 field stimuli and 40 field stimuli conditions, corresponding to data shown in the top panel. For **c** and **d**, arrows indicate field stimulus onset.

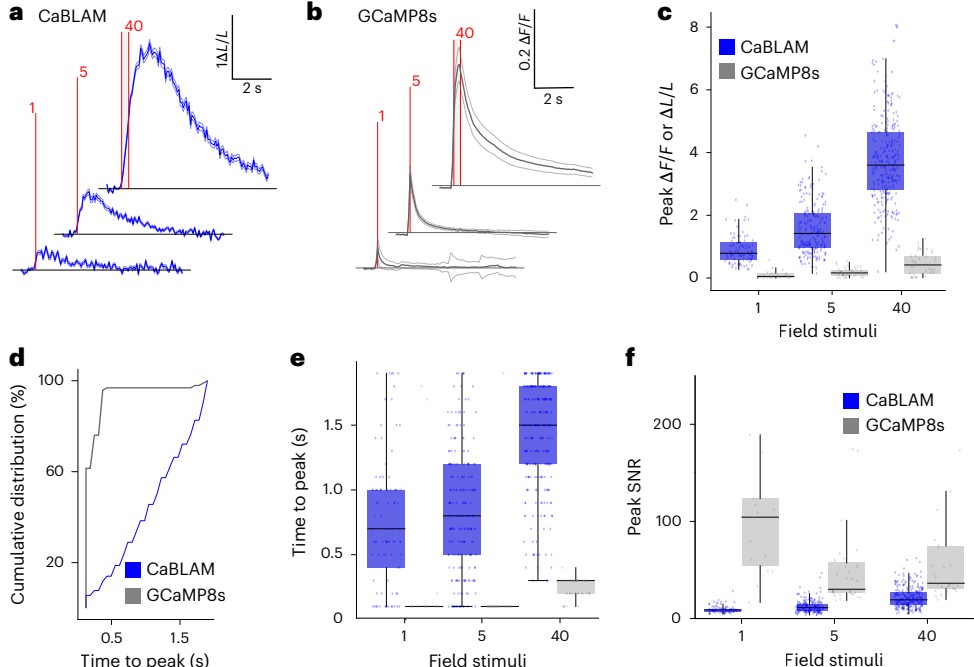

**Fig. 3 | CaBLAM provides high-contrast reporting of stimulus-evoked neural activity in cultured neurons. a**, Bioluminescence $\Delta L/L$ time-locked traces of CaBLAM $Ca^{2+}$ responses to 1, 5 and 40 pulses of 1-ms field stimulations at 83 Hz (red dashed lines indicate field stimulation window). **b**, Same as **a**, but fluorescence $\Delta F/F$ time-locked traces for GCaMP8s. For **a** and **b**, data are shown as mean ± s.e.m. **c**, Peak stimulus-evoked $\Delta F/F$ or $\Delta L/L$ across neurons across increasing field stimulations. **d**, Cumulative distribution of time to peak $\Delta F/F$ or $\Delta L/L$ responses. **e**, Time to peak $\Delta F/F$ or $\Delta L/L$ across neurons in response to increasing field stimulations. **f**, Peak SNR of $\Delta F/F$ or $\Delta L/L$ responses across

neurons at increasing stimulation intensities. For CaBLAM, data were pooled from 7 independent imaging sessions, with the following total number of neurons ($n$) analyzed at the corresponding stimulation numbers (1, 5 and 40): $n$ = 61, 58 and 83. For GCaMP8s, data were pooled from 3 independent imaging sessions, with $n$ = 15, 37 and 38 neurons analyzed at the same respective stimulation numbers. All boxplots show the median, 25th and 75th percentiles (box edges), whiskers extending to the most extreme data points and individual outliers plotted separately.

(Extended Data Fig. 3a) were binned at 10-min intervals, reaching a peak at 30 min, slowly decreasing over the next 2 h, then plateauing at around half the peak brightness in hours 3–5 (Extended Data Fig. 3b). The tactile CaBLAM responses over time were stable, the peak responses within approximately the first hour after FFz administration were smaller than at all later times, and the SNR was not significantly different across time bins (Supplementary Text and Extended Data Fig. 4).

This experiment was concluded while the BL signals and sensory-evoked response were still robust, >5 h after 1 FFz application. This indicates that single, long-duration sessions are possible, and that the duration of CaBLAM imaging is likely limited primarily by the bioavailability of fresh luciferin. After the extended imaging session, in the same mouse, we were able to clearly image neural processes (Extended Data Fig. 5) at a 1 Hz imaging rate at ×40 magnification.

### CaBLAM signals in astrocytes and neurons in zebrafish
We generated a transgenic zebrafish line, *Tg(UAS:CaBLAM)* and a simple microscope to simultaneously measure bioluminescence using a photon-counting PMT and behavior using a machine vision camera. We next used a driver line, *Tg(GLAST)*[33] to express CaBLAM in zebrafish astrocytes (Fig. 5a), and monitored CaBLAM signals associated with spontaneous tail movements. Small, short increases in bioluminescence were associated with smaller movements, and large, prolonged increases followed the largest movements (Fig. 5b). The video frames around the time of large movements (for example, Fig. 5c–e) show that they are composed of strong, asymmetric and uncoordinated tail flicks (Supplementary Video 4). Increases in bioluminescence lasted ~10 s (Fig. 5f), and varied in amplitude (Fig. 5g), and all large movements we observed were accompanied with increases in bioluminescence. Our data suggest that large-amplitude tail movements elicit CaBLAM-derived bioluminescence.

Different genotypes showed similar increases in bioluminescence following the largest movements, but with varied kinetics. Experiments lasted 28–125 min (Extended Data Table 1). *Is(nefma)* labels descending midbrain or hindbrain premotor and spinal cord neurons[34]. CaBLAM-expressing *Is(nefma)* larvae ($n$ = 3) displayed strong increases in bioluminescence with a fast rise followed by a decrease relative to baseline (Fig. 5h). Nonfluorescent siblings ($n$ = 3) ran with identical 1:100 vivazine had baseline counts near the noise floor of the PMT (with infrared illumination on) and showed no detectable fluctuations associated with the largest movements (Fig. 5h). *Tg(GLAST)* larvae with CaBLAM expressed in astrocytes were similar to the example (Fig. 5b–g) with a slower rise time and decrease below baseline (Fig. 5i). *Tg(hcrtr2)* labels both neurons and large vacuolar cells in the notochord[35] and shows a more prolonged response with a comparable decrease below baseline (Fig. 5j). Finally, *Tg(elavl3)* labels most postmitotic neurons[36]; we decreased the concentration of vivazine to 1:1,000 and still observed changes in bioluminescence with a fast rise and minimal decrease below baseline (Fig. 5k). Average BL intensity, number of high-amplitude movements and experiment duration for three head-fixed individuals from each genotype are detailed in Extended Data Table 1. Taken together, our data support the inference that large-amplitude movements are accompanied by prolonged $Ca^{2+}$ flux in both neurons and astrocytes.

We performed two proof-of-principle experiments to explore imaging approaches that preserve spatial information about CaBLAM-derived bioluminescence. First, we used the machine vision camera in our microscope to image a 5 days postfertilization (dpf) *Tg(elavl3)* fish in 1:100 vivazine. A 30 s exposure, with 4 × 4 binning clearly reveals bioluminescence in the brain and spinal cord (Fig. 5l). We could remove the background noise by measuring the pixelwise s.d. across repeated exposures ($n$ = 83, 41.5 min), revealing the slight shifts

in basal tail orientation across this particular experiment (Fig. 5m and Supplementary Video 5). Overlaying an infared-illuminated image of the larva (Fig. 5n) shows the alignment of fish and bioluminescence. We conclude that even a low-cost, uncooled and unintensified camera is sufficient to resolve CaBLAM-derived bioluminescence.

Second, we used an intensified high-speed camera to measure bioluminescence in a head-fixed 4 dpf *Tg(elavl3)* fish in 1:1,000 vivazine (Fig. 5o,p). We could image a head-embedded larva at 20 fps, which was sufficient to resolve large and small amplitude tail movements (Supplementary Video 6). A kymograph of 2 min of imaging the midline of the fish was sufficient to resolve individual swims as the tail moved (Fig. 5q). For one large movement, we observed a strong increase in intensity along the tail; qualitatively, caudal regions stayed bright for longer than rostral regions, consistent with a traveling $Ca^{2+}$ wave. Taken together, our data support the conclusion that CaBLAM-derived bioluminescence can be used to monitor $Ca^{2+}$ flux in neurons and astrocytes in larval zebrafish. Here, we use CaBLAM to reveal intense and prolonged increases in $Ca^{2+}$ flux along the tail following large-amplitude tail movements in head-fixed fish.

## Discussion

CaBLAM is well-positioned to enable activity monitoring across a breadth of applications. The ability to record well-resolved signals from many distinct neurons across hours in head-fixed mice enables robust imaging without the photodamage and bleaching endemic to 1-photon fluorescence imaging. As studies of the neural dynamics supporting complex behavior are being increasingly conducted in free behavioral settings, indicators that do not require fixing the position of the experimental subject and minimize additional implanted hardware are sorely needed. The complexity, weight and size of implanted hardware for imaging in freely moving preparations (for example, GRIN lens-based or fiber photonic) can be cut approximately in half[37] by using a BL GECI in place of a fluorescent GECI. Further, if net activity levels in a given target are the sought after metric, a BL GECI enables dynamic imaging without any implanted systems, as in our zebrafish studies and, in the future, mammalian model systems. Future studies using color-shifted variants of CaBLAM could also allow completely non-invasive imaging of the output of multiple brain areas and/or organ systems during such behavior.

While the SNR advantage afforded by elimination of autofluorescent background partially compensates for the low light output of luciferases in imaging, required exposure times have historically been in the tens of seconds, far from what is necessary to observe meaningful $Ca^{2+}$ dynamics and other fast cellular processes. Imaging intracellular $Ca^{2+}$ at high speeds is possible only when the contrast of an indicator is well above the electronic noise in the camera. BL GECIs derived from GeNL have been imaged at up to 60 Hz in induced pluripotent stem cell-derived cardiomyocytes[17], which display very large changes in cytosolic $Ca^{2+}$ concentration relative to most cells, but displayed a dynamic range of less than twofold despite in vitro characterization suggesting a much higher contrast. We observe that in HeLa cells, displaying smaller swings in $Ca^{2+}$ concentration when responding to histamine stimulation, the high baseline luminescence of the GeNL($Ca^{2+}$) sensor family limits the achievable dynamic range. CaBLAM's low baseline signal prevents a washout of true signal from scattered out-of-focus baseline emission that is common for both fluorescent and BL GECIs in thick tissues. In neurons, CaBLAM reliably reports responses to single action potential stimulation, such as GCaMP8s when imaged at 10 Hz, and can be imaged indefinitely without photobleaching or phototoxicity when provided a steady supply of luciferin substrate. Most importantly, CaBLAM enables imaging of $Ca^{2+}$ flux in awake mice at single neuron resolution with superior SNR to GCaMP6s under 1-photon illumination. Future improvements to CaBLAM's BL brightness are expected to further improve SNR and increase practically achievable frame rates and imaging depths, while future improvements in substrate bioavailability, blood–brain barrier permeability, and stability will continue to simplify substrate delivery and allow fully non-invasive free-behavior experiments.

Phototoxicity and photobleaching are distinct but interrelated phenomena. Exposing cells to low levels of blue light for periods as short as 4 min has been shown to alter their normal physiology[38]. In whole organism imaging, light exposure at typical fluorescent GECI imaging levels can even alter the typical development of nonneural structures, such as bone[39]. Photobleaching is typically approached as a limitation of the sensor that should be minimized for practical purposes. However, phototoxicity can occur even when photobleaching is minimized or eliminated[38,40]. The nature and extent of alterations in cell physiology caused by the illumination source in fluorescent GECI imaging remains minimally documented, particularly in the context of in vivo imaging. Experiments that assess functional changes across several imaging sessions, such as those investigating learning and memory, may be particularly susceptible to these issues, since slow changes in cellular activity due to the actual mechanisms of interest are essentially indistinguishable from changes emerging over the same time scale induced by phototoxicity and photobleaching. Photobleaching in vivo is typically assessed along periods in the tens of minutes, suggesting that longer duration fluorescent GECI imaging experiments (for example, >30 min), investigating changes in cellular dynamics on similar timescales, are necessarily accompanied by phototoxicity. We demonstrate in vivo that imaging across at least 5 h is possible using CaBLAM with stable SNR and very slow degradation in intensity, something that would currently be far beyond what is achievable with any fluorescent GECI.

Translucence makes fish species such as *Danio* and *Danionella* particularly attractive models for BL imaging of $Ca^{2+}$ flux in neurons and glia, but technical challenges have hindered progress. Earlier work

---

**Fig. 4 | CaBLAM shows SNR comparable to GCaMP under epifluorescent illumination during in vivo mammalian imaging. a**, The mouse imaging setup with location of the cranial window and head-fixation on a running wheel. **b**, The CaBLAM and GCaMP6s imaging configurations: CaBLAM was imaged in darkness in a light-tight enclosure after administration of luciferin and GCaMP6s was imaged under epifluorescent illumination. **c**, Mean projection images of an NDNF-Cre CaBLAM and GCaMP6s field of view. **d**, CaBLAM activity traces from 15 neurons in 1 field of view. Vertical gray bars indicate onset of tactile stimuli, and the (bottom) trace is mouse running speed. **e**, Percentages of stimulus responsive cells across animals and sensor groups. Individual mice are represented by dots and group medians by horizontal bars. **f**, Example of 20 tactile stimulation trials for 4 cells from 1 field of view. Gray traces are individual trials, and black traces are the mean. **g**, Mean responses of all positively responsive cells for the NDNF CaBLAM (left) and NDNF GCaMP6s (right) groups sorted in ascending order by latency of the signal at half-maximum. **h**, Mean sensory response across mice in the two groups from positively responsive cells. Stimulation onset is at $t = 0$, indicated by the vertical dashed line. Opaque bars represent the jackknifed 95% CI of the means. Separate left and right $y$ axes for the CaBLAM and GCaMP6s signals, respectively. **i**, Single-trial SNR measurements across all positively responsive cells for the two sensor groups. Dots are individual trials and horizontal lines indicate the median. **j**, Mean running-onset waveforms across animals for the two sensor groups. Running onset is at $t = 0$, indicated by the vertical dashed line. Inset, the average running speed across mice time-locked to onset of running bouts. **k**, Left: single epifluorescence image of a full cranial window expressing CaBLAM pan-neuronally in one animal. Right: the CaBLAM BL average projection image (~4 min recording, $n = 5,119$ images) in the same mouse after retro-orbital injection of CFz. **l**, Mean CaBLAM tactile response ($n = 3$ mice) after retro-orbital injections of CFz (sampled ROI shown by the white circle in **k**). Stimulation onset is at $t = 0$, indicated by the vertical dashed line. For the 2-Hz sampling rate (magenta), opaque shading represents the jackknifed 95% CI of the mean. The darker trace shows an average tactile response from one animal imaged at 10 Hz.

established a first-generation reporter, Aequorin-GFP[10] in zebrafish[12,41], but, in contrast to cardiology[42] and cancer biology[43], BL has rarely been used for $Ca^{2+}$ monitoring in zebrafish neurons, likely due to the low flux produced by Aequorin-GFP in this context and challenges of dealing with off-target (that is muscle) expression[44]. In our experiments, we routinely saw 2–3 orders of magnitude more flux than the $400$–$3,000$ photons $s^{-1}$ in rigorously screened GFP-Aequorin lines[41]. While the lines tested here likely drive broader expression, and we cannot rule out the possibility of low levels of muscle expression, we nonetheless see both strong BL that persists long past the movement

and kinetics that are specific to particular transgenic driver lines. Thus, we view CaBLAM as an enabling tool for future experiments to monitor activity in genetically targeted populations of neurons and glia in freely behaving fish over long time periods.

The scaffold and design principles we have identified in this work will be useful in the design and optimization of bright, high-contrast BL sensors for other biochemical activities as well[45]. In the future, development of red-emitting SSLuc and CaBLAM variants, either through improved red-emitting FRET acceptors (such as optimized long Stokes-shift fluorescent proteins) or engineering of high-activity,

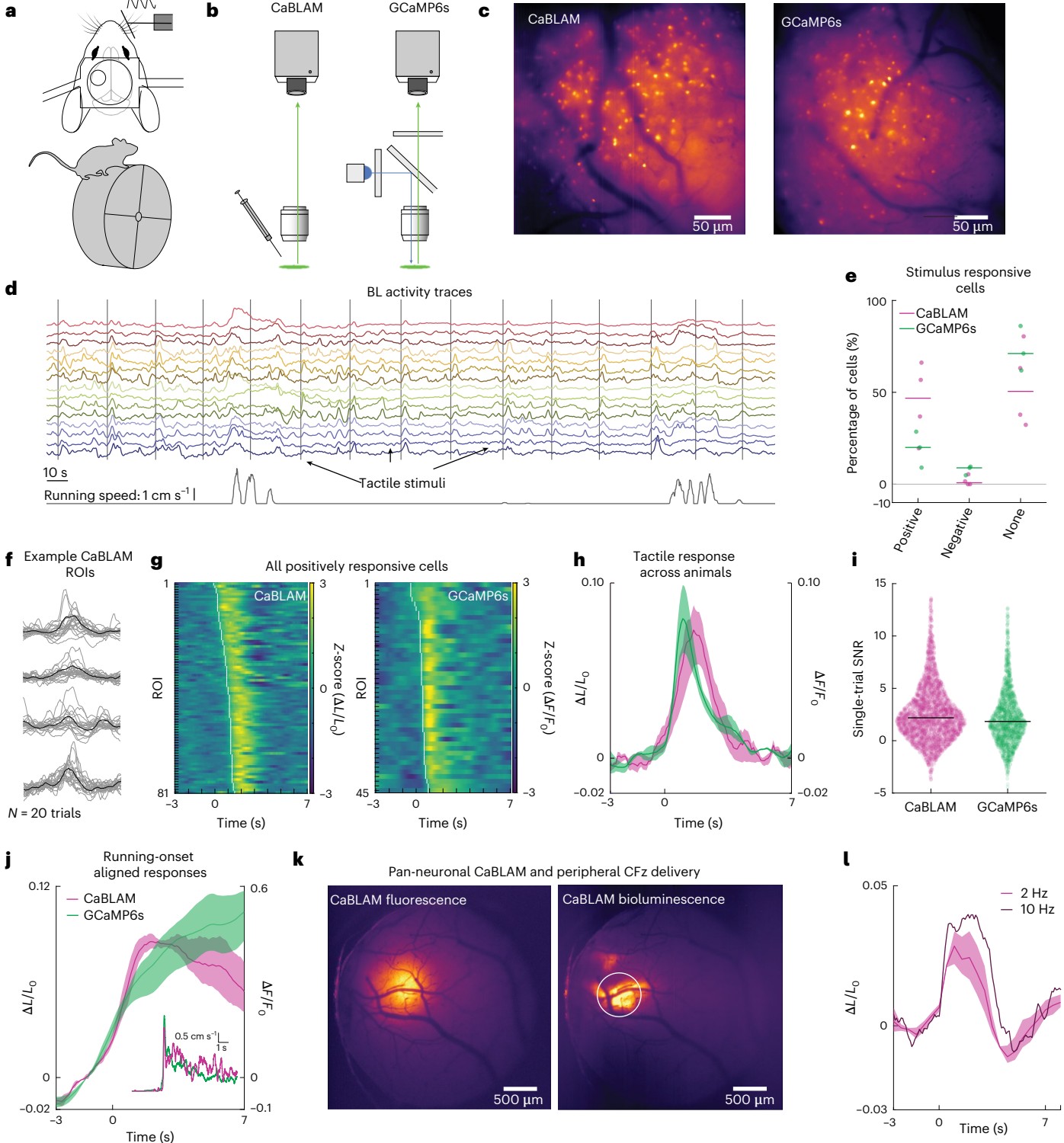

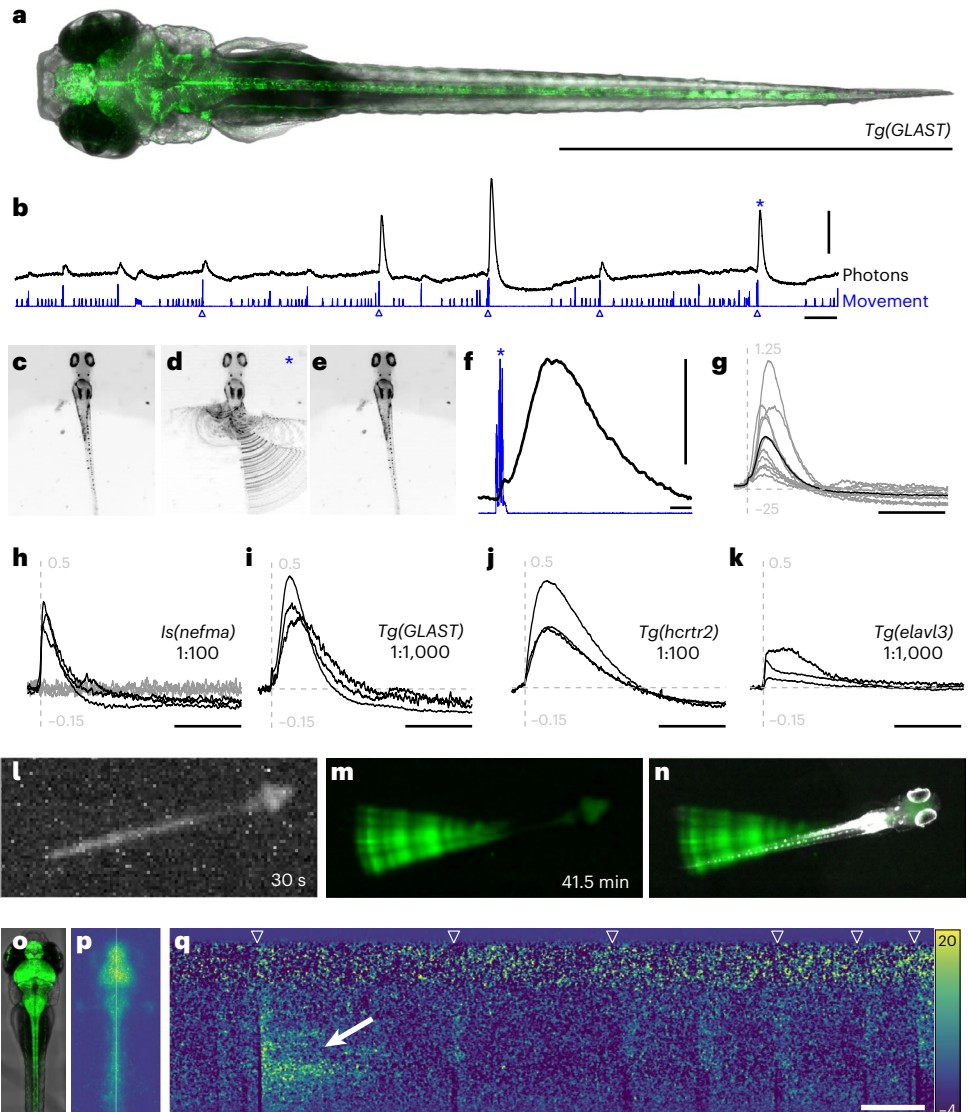

**Fig. 5 | CaBLAM-derived BL signals in astrocytes and neurons follow high-amplitude tail movements in head-embedded larval zebrafish. a**, Confocal image of a 5 dpf *Tg(GLAST)* larva showing expression of CaBLAM in astrocytes (green) and transmitted light (gray). Scale bar: 2 mm. **b**, A representative trace (80 s) of detected photons (black) and movement (blue). Blue triangles correspond to detected high-amplitude movements. Scale bars: 50,000 counts (vertical), 10 s (horizontal). **c**–**e**, Summed frames from a 4 s video corresponding to a high-amplitude movement: 200 ms before the movement (**c**), 400 ms of movement (**d**) and 3,400 ms after the movement (**e**). Changes are only detectable during the movement, marked by strong uncoordinated and asymmetric tail flicks. **f**, A zoomed-in view of photons and movement from **c**–**e**, marked by the asterisk in **b**. Counts begin during the movement and continue for seconds after the movement has ceased. Scale bars: 50,000 counts (vertical), 1 s (horizontal). **g**, Normalized counts during detected high-amplitude movements (gray, *n* = 10) over a full experiment. Black is the mean BL response. Dotted lines at *t* = 0 (vertical) and 0 counts (horizontal), vertical scale is −0.25 to 1.25. Scale bar: 10 s.

**h**–**k**, Normalized counts during high-amplitude movements for four different genotypes. Dotted lines at *t* = 0 (vertical) and 0 counts (horizontal), vertical scale is −0.15 to 0.5. Scale bar: 10 s. **h**, *Is(nefma)* CaBLAM-expressing fish (black, *n* = 3) and negative siblings (gray, *n* = 3) with 1:100 vivazine substrate, 3–4 dpf. **i**, *Tg(GLAST)* at 1:1,000, 3 dpf, *n* = 3. **j**, *Tg(hcrtr2)* at 1:100, 3 dpf, *n* = 3. **k**, *Tg(elavl3)* at a lower dose (1:1,000), 3 dpf, *n* = 3. **l**, A single 30 s exposure of a 5 dpf *Tg(elavl3)* fish at 1:100 vivazine. **m**, Pixelwise s.d. across a stack of 83 30 s exposures. **n**, A composite of the image in **n** and an infrared-illuminated image of the fish. **o**, Maximum intensity projection confocal stack of a 4 dpf *Tg(elavl3)* larva. **p**, Maximum projection of a 2-min intensified image of a 4 dpf *Tg(elavl3)* larva in 1:1,000 vivazine. The vertical white line in the center of the image shows the location of the pixels used for analysis. **q**, A 2-min kymograph of intensity changes over time in the pixels along the white line in **p**. White triangles indicate movements, the white arrow shows the elevated intensity that followed a large tail flip. Scale bar: 10 s.

---

red-emitting substrates, will further enhance the deep-tissue imaging potential of these probes. Another motivation for BL indicators is their potential to be partnered with optogenetic tools, for example to allow simultaneous monitoring of neural activity and light-controlled neural activation, with the lack of fluorescence excitation allowing maximal flexibility in the choice of optogenetic channel. We plan to use CaBLAM to drive optogenetic elements such as channelrhodopsins and light-sensitive transcription factors, making them response to

changes in intracellular $Ca^{2+}$, building on previous work fusing bioluminescence and optogenetics[46–48]. Optimization of CaBLAM kinetics for detection of fast spiking events is another clear target for future development. Continuing efforts also focus on further improving the light production rate of this and other classes of luciferase. Looking forward, if $K_m$ (Michaelis constant) values are maintained at ~10 μM, a diffusion-limited luciferase (that is, with a turnover number, $k_{cat}$, such that $k_{cat}/K_m = 10^8–10^9$ M$^{-1}$ s$^{-1}$) should be capable of generating $10^3–10^4$

photons per second per molecule, which leads us to hope that there is ample room left for future improvements to these enzymes. Given our current observations, efforts to engineer stable caged substrates with oral bioavailability and to increase the delivery rate of these engineered luciferins to the cytoplasm of target cells will be central to realizing the full potential of BL imaging probes.

## Online content

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

[1]Department of Neurosciences, University of California San Diego School of Medicine, La Jolla, CA, USA. [2]Allen Discovery Center for Neurobiology in Changing Environments, University of California San Diego, La Jolla, CA, USA. [3]College of Medicine, Central Michigan University, Mt. Pleasant, MI, USA. [4]Robert J. and Nancy D. Carney Institute for Brain Science and Department of Neuroscience, Brown University, Providence, RI, USA. [5]Department of Neuroscience, New York University Grossman School of Medicine, New York, NY, USA. [6]Department of Otolaryngology, New York University Grossman School of Medicine, New York, NY, USA. [7]University of California San Diego, La Jolla, CA, USA. [8]Tech4Health Institute, New York University Grossman School of Medicine, New York, NY, USA. [9]Department of Biology, University of North Carolina at Chapel Hill, Chapel Hill, NC, USA. [10]Institute of Neuronal Cell Biology, Technical University of Munich, Munich, Germany. [11]National Center for Microscopy and Imaging Research, University of California San Diego, La Jolla, CA, USA. [12]Scintillon Institute, San Diego, CA, USA. [13]Centre for Clinical Brain Sciences, The University of Edinburgh, Edinburgh, UK. [14]Institute for Translational Neuroscience, New York University Grossman School of Medicine, New York, NY, USA. [15]Department of Ophthalmology, New York University Grossman School of Medicine, New York, NY, USA. [16]Department of Pharmacology, University of California San Diego School of Medicine, La Jolla, CA, USA. [17]These authors contributed equally: Gerard G. Lambert, Emmanuel L. Crespo, Jeremy Murphy. ✉e-mail: ncshaner@health.ucsd.edu

## Methods

### General molecular biology

*E. coli* strain NEB 10-beta (New England Biolabs, cat. no. C3020K) was used for all cloning, library screening and recombinant protein expression. PCRs were performed using Phusion DNA polymerase (New England Biolabs) and the supplied 'GC buffer' under manufacturer-recommended conditions. Custom DNA oligonucleotide primers were purchased from IDT. Restriction enzymes were purchased from New England Biolabs and restriction digests followed the manufacturer's protocols. All plasmids were constructed using Gibson assembly[49] of PCR or restriction digest derived linear DNA fragments[21,50]. NEB 10-beta *E. coli* were electroporated following the manufacturer's protocol. Unless otherwise noted below, chemicals were purchased from Sigma-Aldrich.

### Construction of *E. coli* expression plasmids and libraries

All *E. coli* expression plasmids were constructed using the pNCST backbone[21,50], which allows induction-free expression in most common strains. A linear DNA fragment of the pNCST vector was generated by PCR (primers, Supplementary Data 2). Designed luciferase and new GECI coding sequences were initially prepared as synthetic genes (IDT), as were many designed variants during directed evolution in cases where site-directed mutagenesis would be impractical. Mammalian expression plasmids encoding GeNL(Ca²⁺)_480 (ref. 17) (plasmid no. 85205) and CaMBI[18] (plasmid no. 124094) were obtained from AddGene and amplified by PCR (primers, Supplementary Data 2) for insertion into the pNCST expression vector. Error-prone mutagenesis was performed using the GeneMorph II kit (Agilent Technologies)[21] with a target mutation rate of ~5 base changes per kilobase of amplified product. Site-directed libraries were constructed by Gibson assembly of PCR fragments amplified to introduce mixed-base codons[21]. Selected clones from library screening were sequenced using standard Sanger sequencing (Eton Biosciences). Libraries were introduced to *E. coli* via electroporation with plating at densities between $10^3$ and $10^5$ colonies per 10 cm petri dish of LB/Agar supplemented with 100 μg ml⁻¹ carbenicillin (Fisher BioReagents) and incubated overnight at 37 °C. All plasmid sequences are provided in Supplementary Data 1.

### Luciferin substrates

All substrates and stock solutions were stored at −80 °C. Stock solutions of native CTZ (NanoLight, cat. no. 303) were prepared by dissolving in NanoFuel Solvent (NanoLight, cat. no. 399) to a final concentration of 10 mg ml⁻¹ (23.6 mM). Fz stock solution is a component of the Nano-Glo Luciferase Assay System (Promega, cat. no. N1110 or N1120). Because Promega does not disclose the Fz concentration in this stock solution, we measured absorbance spectra of samples of the stock solution diluted in methanol, then calculated the concentration of the stock solution using the extinction coefficient of Fz (21,000 M⁻¹ cm⁻¹ at 254 nm) from published patent application US2020/0109146A1, giving a concentration of 4.6 mM for the Promega stock solution. From this, we calculated the final Fz concentrations from each dilution ratio used (typically a 1:500 or 1:1,000 dilution of the Fz stock solution for live imaging and 1:100 for in vitro assays) and provide this value in the text and figures.

Water soluble hCTZ (Nanolight, cat. no. 3011) was dissolved in sterile saline (4.55 μg μl⁻¹). Nano-Glo Fluorofurimazine In Vivo Substrate (FFz, Promega, cat. no. N4110) was dissolved in 525 μl of phosphate buffered saline per vial (4.6 μmol per 525 μl). Nano-Glo Cephalofurimazine In Vivo Brain Substrate (CFz9, Promega, cat. no. CS3553A01) was dissolved in 1 ml of 0.2 M Tris buffer (pH 8.0) per vial (4.2 μmol per 1 ml). As needed for FFz and CFz, the lyophilized solids were broken up into aliquots and stored at −80 °C; for use they were dissolved at the concentrations detailed above. For all live imaging experiments, luciferase substrates were dissolved just before use.

The caged Fz derivative Nano-Glo Vivazine Live Cell Substrate (Promega, cat. no. N2580) was diluted 1:100 or 1:1,000 into aquarium water immediately before pre-incubation of zebrafish larvae (described below). We did not attempt to measure the stock concentration of vivazine, and report concentrations as dilution factors.

### Luciferase library screening

Error-prone mutagenesis and site-directed libraries of luciferases were screened by spraying a sterile solution of 50 μM CTZ or 46 μM Fz (1:100) in 50 mM Tris-HCl (pH 7.5) onto individual plates using a small atomizer. For 'manual' screening, the entire screening process was performed in complete darkness or very dim far-red LED light, with care taken to dark-adapt vision before starting. Sprayed plates were examined by eye, and the brightest colonies were immediately marked. For image-based screening, plates were placed immediately into a light-tight chamber and imaged with a PRIME 95b sCMOS camera (Photometrics), followed by identification of the brightest colonies using ImageJ and manual marking on the plates. Marked colonies were grown individually in 5 ml of 2× YT medium supplemented with 100 μg ml⁻¹ carbenicillin with shaking at 250 rpm at 37 °C overnight. Colony screening plate densities (typically $10^3$–$10^5$ colonies per plate) were determined empirically for each mutagenesis round to balance coverage of library diversity with reliable detection of bright variants. No statistical calculation was performed; throughput was consistent with previous directed-evolution screens.

### Protein expression, purification and in vitro experiments

The pNCST plasmid is useful for protein library screening and expression because it does not require an inducer in most strains of *E. coli*. For characterization of selected luciferase and GECI clones, 1 ml of each overnight liquid culture was pelleted by centrifugation and the supernatant was discarded. The bacteria were then resuspended once in ultrapure water to remove residual culture medium and pelleted again by centrifugation, discarding the supernatant. The washed pellet was then resuspended in 400 μl of B-PER (Thermo) reagent and incubated at room temperature for 20 min with gentle rocking to extract soluble proteins, followed by a final centrifugation to pellet insoluble components and yield a clear lysate. Clarified B-PER lysates were used directly for characterization experiments in most cases, since we observed rapid degradation of some proteins when subjected to additional purification procedures. In all cases, proteins prepared as clarified B-PER lysates were assayed within 6 h of extraction.

For experiments using purified recombinant protein, B-PER lysates were bound to equilibrated cobalt beads[21,50], washed with 50 mM Tris-HCl (pH 7.5), 150 mM NaCl, 5 mM imidazole and eluted in 50 mM Tris-HCl (pH 7.5), 150 mM NaCl, 500 mM imidazole, then buffer-exchanged into 50 mM Tris-HCl (pH 7.5), 150 mM NaCl using Zeba desalting columns (Pierce). Purified proteins were stored at 4 °C.

For brightness comparisons in vitro, B-PER lysates of mNG-fused luciferase clones were diluted in 50 mM Tris-HCl (pH 7.5), 150 mM NaCl to a peak mNG absorbance value <0.05 and 200 μl of samples were loaded into a clear-bottom black 96-well plate (Corning) along with B-PER lysates from 1 or more controls (for example, GeNL). Fluorescence emission spectra were recorded for each sample with excitation at 480 nm (5-nm bandwidth) using a Tecan M1000Pro Infinite multimode plate reader. To normalize enzyme concentrations based on the fluorescence, a dilution factor was calculated for each well based on the peak fluorescence emission. Samples were then diluted into new wells to a total volume of 50–100 μl and were measured again to verify that fluorescence intensities varied by ≤5% over all wells. The plate was then transferred to a ClarioStar multimode plate reader (BMG) to measure luminescence kinetics. Luciferin solutions at 50 μM (CTZ) or 46 μM (Fz, 1:100) were prepared in the same Tris buffer as described above and injected in volumes of 100–150 μl individually into wells with concurrent measurement of luminescence intensity

for a total measurement time of 50 s per well. These sample and injection volumes were chosen to reduce variability in measurements due to uneven mixing, which can occur if the injection volume is smaller than the sample volume. Peak and steady-state intensities were then normalized by dividing by the previously measured fluorescence of each well to account for remaining small variations in concentration between wells, and these normalized values were used for brightness comparisons between clones.

## Ca²⁺ titrations in vitro

B-PER lysates of mNG-fused GECIs were diluted ~100-fold in 30 mM 3-($N$-morpholino)propanesulfonic acid (MOPS) (pH 7.2), 100 mM KCl, 1 mM $MgCl_2$ and then dispensed into 3 rows (36 wells) of a 96-well plate, 5 µl per well. EGTA-buffered $Ca^{2+}$ solutions with free $Ca^{2+}$ concentrations ranging between 0 and 39 µM were then prepared from commercial stock solutions (Invitrogen cat. no. C3008MP), also in 30 mM MOPS (pH 7.2), 100 mM KCl and supplemented with $MgCl_2$ to a final concentration of 1 mM. The presence of $Mg^{2+}$ provides more physiologically relevant conditions for this assay, but slightly alters free $Ca^{2+}$ concentrations in EGTA-based buffers, so we calculated final free $Ca^{2+}$ for each condition using parameters determined in ref. 51 and validated these calculations using the small-molecule $Ca^{2+}$ dye fluo-4 (ThermoFisher cat. no. F14200). To obtain a fuller titration curve when titrating the low-affinity CaBLAM_294W variant, we prepared an additional unbuffered solution of 100 µM $CaCl_2$ in 30 mM MOPS (pH 7.2), 100 mM KCl, 1 mM $MgCl_2$.

For the titration, 100 µl of $Ca^{2+}$ buffer solution was added to each well containing sample to produce three or more identical rows of the same concentration series. A solution of 46 µM Fz (1:100) in the same $Mg^{2+}$-containing MOPS buffer was prepared and injected in volumes of 150 µl individually into wells with concurrent measurement of luminescence intensity for a total time of 20 s per well. Titration series comprised six independent measurements for CaBLAM and comparable numbers for other indicators (Fig. 1c and Table 1). No statistical method was used to predetermine sample size; replicate numbers were based on previous experience and consistency with similar sensor characterization studies. The area-under-the-curve (AUC) luminescence values were calculated for each kinetic curve and normalized per replicate series. Custom Python code was then used to fit titration curves using a three-parameter Hill equation: $y = y_{min} + (1 - y_{min})/\left(1 + (EC_{50}/x)^{n_H}\right)$, where $y_{min}$ is the minimum normalized emission, $EC_{50}$ is the $Ca^{2+}$ concentration at half-maximal response and $n_H$ is the Hill coefficient. Confidence intervals were calculated using the full covariance matrix to account for parameter correlations. Contrast ratios were calculated by taking the reciprocal of the mean of all data points at $Ca^{2+}$ of <10 nM, after normalization, for all sensors except CaBLAM_294W, for which we used all points at $Ca^{2+}$ of <50 nM. Plots were generated using custom Python scripts with matplotlib. Marker transparency was automatically adjusted based on local density to improve visibility of overlapping data points. Color schemes were selected to maximize contrast and accessibility. All plots use logarithmic $Ca^{2+}$ concentration scales and linear normalized emission scales (0–1). No data were excluded from this analysis.

## Construction of mammalian expression plasmids

Mammalian expression constructs were generated by Gibson assembly of PCR-amplified coding sequences (primers, Supplementary Data 2) into restriction-digested pC1 (CMV promoter, modified from pEGFP-C1, Clontech) or pCAG (CMV enhancer fused to chicken beta-actin promoter) vector fragments. When vector fragment preparation by restriction digest was impractical, we PCR-amplified vector fragments instead (primers, Supplementary Data 2). We observed qualitatively less variable and longer-lasting expression using the pCAG vector in most cases. Both vectors were suitable for all constructs tested. Mammalian expression plasmids encoding GeNL(Ca²⁺)_480 (ref. 17)

(plasmid no. 85205) and CaMBI[18] (plasmid no. 124094) were obtained from AddGene. The plasmid pCAG_EGFP_GPI (plasmid no. 32601) was obtained from AddGene and digested with $Sac$I and $Xho$I to generate a vector fragment for insertion of GeNL and GeNL_SS for GPI-anchored extracellular display. For the Fluoppi assay[52], pCMV-PB1-AG (plasmid no. 178861) was obtained from AddGene and digested with $Bam$HI and $Kpn$I to generate a vector fragment for insertion of NanoLuc and SSLuc followed by a T2A peptide and the red fluorescent protein mCherry[53] as a transfection marker (primers, Supplementary Data 2).

## Cell line culture, transfection and imaging

HeLa (cat. no. CCL-2) and U2OS (cat. no. HTB-96) cell lines were purchased from ATCC. N2a cells (Neuro 2a, ATCC, cat. no. CCL-131) were a gift from D. Black (UC Los Angeles). Cells were maintained under standard culture conditions with incubation at 37 °C and 5% $CO_2$. Growth medium for HeLa cells was high-glucose DMEM (Gibco) supplemented with 10% fetal bovine serum (FBS) (Gibco), for U2OS was McCoy's 5a (Gibco) with 10% FBS, and for N2a was Eagle's Minimum Essential Medium (Gibco) with 10% FBS. For transfection and imaging, cells were plated at a density of $1–2 \times 10^5$ cells per ml on coverslip-bottom 35 mm dishes (Mattek) or 13 mm round coverslips (BioscienceTools) placed in six-well culture plates and incubated overnight. The following day, cells were transfected using polyethyleneimine[50]. Cells were imaged 24–48 h posttransfection. Analyses included multiple fields and dozens of transfected cells per condition, as shown for CaBLAM ($n = 61–83$ neurons) and GCaMP8s ($n = 15–38$ neurons) across independent sessions (Figs. 2 and 3). Sample sizes were not predetermined statistically but were chosen to ensure reproducibility of mean $\Delta L/L_0$ and $\Delta F/F_0$ values across replicate wells and imaging sessions.

Before imaging nonsensor luciferase constructs and fusions, cells were gently rinsed with fresh medium, leaving 500 µl of medium in each dish for Mattek dishes. For cells plated on coverslips, slips were transferred to a low-profile coverslip chamber (BioscienceTools) with a silicone gasket and overlaid with 500 µl of medium. In either case, cells were transferred to the stage-top environmental chamber and allowed to equilibrate for at least 5 min before imaging.

Image acquisition was performed in a stage-top environmental enclosure (37 °C, 5% $CO_2$; Okolab) on a Nikon Ti-E microscope and an Andor iXon Ultra 888 EMCCD camera. Cells expressing BL constructs were imaged with a Plan Apo λ ×20 Ph2 DM 0.75 numerical aperture (NA) objective (Nikon). For reference fluorescent images in constructs containing mNeonGreen, green fluorescence was excited with a Spectra X LED source (Lumencor) using the 475/28 nm channel and a FF495-Di03 dichroic (Semrock), and emission was selected with a FF01-520/35 filter (Semrock). The camera was set to 30 MHz at 16-bit horizonal readout rate, 2× pre-amplifier gain and an electron multiplication gain setting of 3 for fluorescence acquisition. For BL signal recording, the illumination source was turned off and emission was collected either with no filter to maximize light collection efficiency or with the FF01-520/35 filter to minimize background light leakage. The camera was set to 30 MHz at 16-bit horizonal readout rate, 2× pre-amplifier gain and electron multiplication gain of 300, with either $1 \times 1$ or $2 \times 2$ binning. Exposure times were set to between 50 and 500 ms for bioluminescence time series and between 500 ms and 1 s for single images.

For $Ca^{2+}$ indicator time series imaging, cells were only plated in Mattek dishes and were not rinsed, a handling method we found prevented induction of extraneous $Ca^{2+}$ signals in the cytosol. Medium was carefully removed to leave 500 µl volume remaining in each dish, and 500 µl of medium from the dish was reserved and used to dilute the Fz substrate to eliminate small changes in ionic strength and osmolarity that can arise due to evaporation during culture. Baseline images were collected for at least 30 s in the dark before luciferin addition. Fz diluted in culture medium was then injected to a final concentration of 4.6 µM (1:1,000) and the BL signal was recorded for 60 s. Next, 50 µl of concentrated ionomycin was injected to give a final concentration of 20 µM

and images were recorded for an additional 60–120 s for determination of the maximal contrast in cells. The concentration of $Ca^{2+}$ in the culture medium was ~1 mM (but likely buffered to some degree by other medium components) and therefore expected to produce a 'maximal' physiological $Ca^{2+}$ concentration in the cytosol in the presence of ionomycin without additional supplementation. For imaging induced $Ca^{2+}$ oscillations, ionomycin injection was replaced by injection of 50 µl of concentrated L-histamine to give a final concentration of 50 µM, and images were recorded for an additional 10–20 min.

## Fluoppi oligomerization assay

To determine the oligomeric state of SSLuc when expressed in mammalian cells, we adapted the Fluoppi[52] scaffold-assembly assay to a BL readout by fusing the PB1 oligomerization domain to each luciferase variant (PB1-NanoLuc, PB1-SSLuc). HeLa cells transfected with each plasmid were imaged 24 h posttransfection as described above. In the Fluoppi framework, PB1-driven higher-order assembly yields intracellular puncta, and diffuse signal indicates absence of assembly. Raw images were background-corrected with the rolling ball algorithm (150-pixel radius), followed by $\log_{10}$ transformation to reduce dynamic range and linear rescaling to the full range of the transformed data before scoring. For each construct, expressing cells were manually scored as puncta-free ('good') versus ≥1 punctum ('bad'), blinded to condition. All data were included in the analysis. Counts from 2 independent wells were pooled per construct for the primary 2 × 2 comparison and significance was assessed by a 2-sided Fisher's exact test.

## Primary rat cortical neuron culture, transfection and $Ca^{2+}$ imaging

Primary cortical neurons were prepared from 1 postnatal day 2 Sprague–Dawley rat pup (Charles River Laboratories) of undetermined sex. Animal procedures were approved by the Institutional Animal Care and Use Committee of UC San Diego. Rat pups were housed with the dam in a vivarium on a 12-h reversed light–dark cycle and had free access to food and water. Cortex was dissected out and neurons were dissociated using papain[54]. Transfection of neuronal cells with pCAG-CaBLAM and pCAG-GeNL(Ca2+)_480 was done by electroporation using an Amaxa Nucleofection Device (Lonza) at day-in-vitro 0 (DIV0). Neurons were cultured on poly-D-lysine coated 35 mm coverslip-bottom dishes (Mattek) in Neurobasal A medium (Life Technologies) supplemented with 1× B27 Supplements (Life Technologies), 2 mM GlutaMAX (Life Technologies), 20 U ml$^{-1}$ penicillin and 50 mg ml$^{-1}$ streptomycin (Life Technologies) for 2–3 weeks before imaging, refreshing half of the medium every 2–3 days.

For $Ca^{2+}$ imaging with KCl-mediated depolarization, Mattek dishes were treated similarly to cell lines, with removal of all but 500 µl of medium, equilibration in the stage-top incubation chamber on the microscope and imaging following the same basic protocol as described above for ionomycin and L-histamine experiments. Dark images were recorded for ~30 s, followed by injection of Fz diluted in 1,000 µl of medium at a final concentration of 4.6 µM (1:1,000) and continuous imaging for 60 s. Finally, all neurons were rapidly depolarized by injection of 500 µl of medium supplemented with 120 mM KCl to give a final concentration of 30 mM KCl and images were collected for an additional 30–60 s.

## Primary rat hippocampal neuron culture and $Ca^{2+}$ imaging

Primary E18 rat hippocampal neurons were prepared from tissue shipped from BrainBits (Transnetyx) following the vendor's protocol. Neurons were seeded on poly-D-lysine-coated 18 mm glass coverslips (Neuvitro) in 12-well tissue culture plates ($1 \times 10^5$ neurons per well) and grown in Gibco Neurobasal Media supplemented with 2% B27, 0.1% gentamycin and 1% GlutaMAX (all from Invitrogen). The next day, DIV1, neurons were transduced with AAV9-Syn ($5 \times 10^9$ gc per well) encoding jGCaMP8s (AddGene 162374) or CaBLAM (in-house

prep). AAV9-Syn-CaBLAM was generated by triple lipofection of HEK293-FT cells and harvesting viral particles[55]. Neurons were imaged between DIV 20 and 30.

For $Ca^{2+}$ imaging with electrical stimulation, coverslips were washed three times and imaged in artificial cerebrospinal fluid (ACSF), containing: 121 mM NaCl, 1.25 mM $NaH_2PO_4$, 26 mM $NaHCO_3$, 2.8 mM KCl, 15 mM D(+)-glucose, 2 mM $CaCl_2$, 2 mM $MgCl_2$ (maintained at ~37 °C, pH 7.3–7.4, continuously bubbled with 95% $O_2$/5% $CO_2$ vol/vol). All imaging was conducted in the RC-49MFSH heated perfusion chamber where the temperature was continuously monitored (Warner Instruments), and only in cultures where neurons were evenly distributed across the entire coverslip. In addition, only coverslips with minimal neuronal clumping or astrocytic growth were included for $Ca^{2+}$ imaging. A Teensy 3.2 microcontroller was used to synchronize the camera frame number with electrical stimulation using BNC cables connected to EMCCD camera and the SIU-102 stimulation isolation unit (Waner Instruments). Custom written scripts were developed to generate analog signals for precise control current stimulation[56]. Briefly, current field stimulation with a 1 ms pulse width at 40 mA and 83 Hz was used. The length of time for each stimulation period was varied (12 ms, 36 ms, 60 ms, 120 ms, 240 ms, 960 ms, 1,920 ms) to achieve the corresponding number of action potentials (1, 3, 5, 10, 20, 80 and 160 action potentials).

For fluorescent and BL $Ca^{2+}$ imaging, a ×20 objective lens (0.75 NA) and iXon Ultra 888 EMCCD (Andor Technology) camera were used. Fz (Promega, cat. no. N1120) was diluted at 1:1,000 or 1:500 in bubbled ACSF to give final concentrations of 4.6 µM or 9.2 µM, both of which provided a sufficient BL signal for imaging at 10 Hz. Image acquisition parameters were as follows: 0.09 s exposure time, 4.33-µs vertical pixel shift, normal vertical clock voltage amplitude, 10 MHz at 16-bit horizonal readout rate, 1× pre-amplifier gain and 2 × 2 binning. The electron multiplication gain was set to 300 for BL image acquisition and it was not enabled during fluorescent imaging. ACSF was continuously perfused in the heated chamber before imaging and during bright field and fluorescence imaging to determine a field of view before beginning BL $Ca^{2+}$ imaging. During BL $Ca^{2+}$ imaging, fresh ACSF was used to dilute the Fz and kept at 37 °C in a water bath. BL imaging included a 60 s initial period of ACSF perfusion without Fz followed with ACSF/Fz at the designated concentration for the remainder of the imaging session. Fz concentration in the imaging chamber typically reached equilibrium within 60 s after initiating ACSF/Fz perfusion, as judged by baseline BL signal in the neuron cell bodies.

For electrical field stimulation experiments, responses were analyzed from 20–276 CaBLAM-expressing neurons and 15–38 GCaMP8s-expressing neurons pooled from three to seven independent sessions (Fig. 3). These sample sizes were guided by established indicator benchmarking studies and were sufficient to detect significant differences in amplitude, latency and SNR distributions.

## Image processing and data analysis

All image processing was performed using ImageJ (versions 1.54a to 1.54p). It is important to note that processing of bioluminescence imaging data requires compensation for the average dark pixel offset arising from the detector's dark current characteristics and by small light leaks in the optical path. For each acquisition setting, dark time series were acquired under identical conditions but without illumination or addition of luciferin substrates. Dark time series with at least 60 frames were used to generate single-frame 'dark' images for each acquisition setting, containing the median value for each pixel over the time series. For single bioluminescence images of fusion proteins, the appropriate dark image was subtracted from the raw image. For time series, the appropriate dark image was subtracted from each frame of the time series. These dark-subtracted time series were summed using the 'Z project' function in ImageJ to create low-noise images for generation of ROIs for analysis. Further settings used for image display

(for example, thresholding, output scaling, lookup tables and so on) are described for individual images in their corresponding figure legends.

To extract data from time series for downstream analysis, ROIs corresponding to individual whole cells were identified from summed time series images using the Cellpose v.3.0.10 (ref. 57) 'cyto' model followed by manual curation. An additional 'background' ROI was drawn manually for each time series in an area devoid of cells. Mean intensities were then measured for each ROI at each time point from the dark-subtracted time series and data transferred to Excel for subsequent steps. The 'background' ROI value was subtracted from each cell ROI at each time point to account for any variable background signal arising from changes in light leakage (for example, when the injection port is uncovered) as well as any bioluminescence generated by extracellular luciferase or GECI molecules that escape from damaged, dead or lysed cells. Note that in cases where cells can be rinsed, the extracellular bioluminescence signal is typically negligible, but baseline shifts from changes in light leakage are inevitable, in our experience, making this step critical for accurate quantitation. For each ROI (whole cell) in a time series, we also calculate the mean dark value during the first 30 s before Fz injection and subtract this value from all time points to bring the dark baseline as close to zero as possible, compensating for any remaining offset in the data not captured by previous processing steps.

For $Ca^{2+}$ indicators, the baseline luminescence emission intensity was determined for each cell by taking the average value of time points in the steady-state phase after initial Fz injection but before ionomycin or L-histamine injection, typically observed between 30 and 40 s after Fz injection (following a small transient cytosolic $[Ca^{2+}]$ increase) in most cells. The indicator signal at each time point was then calculated as the change in luminescence relative to the baseline luminescence ($\Delta L/L_0$) for each cell. No data were excluded from this analysis.

### Dose–response determination in cultured cells
**Luminescence dose–response assay and AUC quantification.** N2a (Neuro 2a) cells (P30-P45) were plated $3 \times 10^5$ cells per well in a 6-well plate and transfected with 2 µg of pcDNA3-CMV-CaBLAM using Lipofectamine 2000 (Invitrogen cat. no. 11668027). 48 h later, cells were gathered, resuspended in external solution (135 mM NaCl, 10 mM HEPES, 2 mM $CaCl_2$, 2 mM KCl) and distributed in equal volumes to a 96-well plate. Cephalofurimazine (CFz9, Promega cat. no. CS3553A01) was kept on dry ice until immediately before use and resuspended in 1 ml of 0.2 M Tris according to the manufacturer's instructions. FFz (Promega cat. no. N4110) was resuspended in 525 µl of sterile phosphate buffered saline, aliquoted and placed in −80 °C until use. Fz (Promega cat. no. N1150) was kept on ice until use. A second 96-well plate was prepared with dilutions of CFz, FFz and Fz, and substrates were transferred to the plate containing cells immediately before luminescence measurements. Luminescence was quantified with a Synergy HTX Multi-Mode Microplate Reader (BioTek) and Gen5 v.3.11 acquisition software using a top-positioned optic, 135 gain, 0.01 s integration time and 1 mm read height at 33 s intervals for 20 min. Dose–response experiments were performed using multiple wells per concentration to derive $EC_{50}$ and Hill parameters with $R^2 > 0.95$ (Supplementary Figs. 7 and 8). No formal sample-size calculation was performed; replicate numbers were selected to ensure smooth curves and reproducible AUC measurements across independent runs.

### Dose–response curve fitting and plotting.
Microplate reader data were imported from raw CSV files, and the AUC was computed for each well using the trapezoidal rule using custom Python code (Python v.3.11.7). Each CSV file included a row specifying substrate concentrations and a column for acquisition time in minutes. To visualize the dose–response curves, AUC values were grouped by concentration, and the mean ± standard error of the mean (s.e.m.) was calculated. For each substrate, either a three-parameter Hill function or a hybrid model

combining a Hill function with a linear decay term was fitted to the mean response values. Curve fitting was performed using non-linear least squares optimization (scipy.optimize.curve_fit) (scipy v.1.11.4 and matplotlib v.3.8.0), with bounds applied to constrain biologically implausible parameter estimates. No data were excluded from this analysis.

The Hill-only model took the form:

$$y = \frac{\text{top}}{1 + \left(\frac{EC_{50}}{x}\right)^n}$$

where $n$ is the Hill slope.

While the hybrid model added a linear decay term:

$$y = \left[\frac{\text{top}}{1 + \left(\frac{EC_{50}}{x}\right)^n}\right] + (mx + c)$$

where $m$ is the slope of the decay and $c$ is the offset. Goodness-of-fit was quantified using the coefficient of determination ($R^2$).

**Bioluminescence imaging.** N2a (Neuro 2a) cells (P30-P45) were plated on 18-mm coverslips coated with 0.1 mg ml$^{-1}$ poly-D-lysine (Gibco A389040) at $1 \times 10^5$ cells per dish in 35-mm dishes in DMEM + 1% FBS. Cells were transfected with 0.25 µg CaBLAM using Lipofectamine 2000 (Invitrogen no. 11668027). Then 24 h after transfection, media was changed to DMEM + 1% FBS + 20 µM retinoic acid (Sigma no. R2625). 24 h later, cells were imaged with an iXon Ultra 888 EMCCD camera mounted on an Eclipse FN1 microscope (Nikon) with a ×16 immersion objective (0.8 NA, WD 3.0, Nikon no. MRP07220). Coverslips were rinsed with external solution and placed on a Quick Change Imaging Chamber (Warner Instruments, no. RC-41LP) with 500 µl of external solution. A reference fluorescence image was taken before bioluminescence acquisition at 30 MHz at 16-bit horizontal readout rate, 1× pre-amplifier gain, electron multiplication gain of 3 and exposure time of 0.1 s. Bioluminescence time series were acquired with Andor Solis 64 bit v.4.32 at 10 Hz with the camera set to 0.09 s exposure time, 4.33-µs vertical pixel shift, normal vertical clock voltage amplitude, 10 MHz at 16-bit horizonal readout rate, 1× pre-amplifier gain and 2 × 2 binning. Images were collected for 1 min in the dark before adding luciferin. Luciferins were diluted in external solution and added to the coverslips to reach the final concentrations described in the figures. After 10 min, ionomycin was added to a final concentration of 2 µM and images were collected for another 30 min.

**Data processing and analysis.** Image stacks were processed using ImageJ as described above and data was exported from ImageJ as .csv files. The rest of the analysis was completed using custom Python code (Python v.3.11.7, scipy v.1.11.4 and matplotlib v.3.8.0) Traces from individual cells were first smoothed using a Bessel filter (fourth order, cutoff 0.1 Hz). The baseline luminescence ($L_0$) was calculated for each cell by taking the mean value of time points in the steady-state after luciferin injection, which was typically 9.3–9.4 min after luciferin injection. The maximum luminescence ($L_{max}$) was calculated as the peak luminescence after ionomycin injection. Both $L_0$ and $L_{max}$ values were normalized to the mNeonGreen fluorescence values collected from the reference fluorescence image for each ROI. No data were excluded from this analysis.

### In vivo $Ca^{2+}$ imaging in mice
**Animals.** Mice were housed in a vivarium on a 12-h reversed light–dark cycle and had free access to food and water. All procedures were conducted in accordance with the guidelines of the National Institute of Health and with approval of the Animal Care and Use Committee of Brown University.

Seven heterozygous NDNF-Cre mice (2 female/5 male; 23–35 weeks old on imaging day; JAX stock no. 030757) were used for in vivo imaging to selectively express CaBLAM or GCaMP6s in NDNF expressing cortical layer 1 interneurons[58]. Three additional heterozygous NDNF-Cre mice (0 female/3 male, each 14 weeks old on imaging day) were injected with a pan-neuronal CaBLAM to test peripheral luciferin delivery.

**Surgical procedures.** For the infusion experiments, four mice were injected with an adeno-associated virus (AAV) vector encoding a floxed version of the CaBLAM sensor (AAV9-ef1a-DIO-CaBLAM) and another three mice were injected with a floxed version GCaMP6s (AAV2/1-CAG-FLEX-GCAMP6s). Animals were assigned randomly to each group. For the peripheral luciferin delivery experiments, three mice were injected with an AAV encoding CaBLAM pan-neuronally (AAV2/9-hsyn-CaBLAM). Each animal was anesthetized (1–2% isoflurane), fitted with a steel headpost and injected with viral constructs in a 3 mm craniotomy centered over left somatosensory barrel cortex (−1.25 anteroposterior, 3.5 mediolateral relative to bregma).

Viral injections were performed through a glass pipette in a motorized injector (Stoelting Quintessential Stereotaxic Injector, QSI). A glass window was then placed over the open craniotomy and cemented with dental cement (C & B Metabond). Mice received a single injection of a given construct at a location within 1 mm of the center of our somatosensory coordinates at depths of 500 µm. Each injection was 500 nl in volume and delivered at a rate of 100 nl min⁻¹. The glass injection pipette was then allowed to rest for an additional 10 min.

Mice receiving direct cortical infusions of luciferin were fitted with a cannula (Plastics One, C315DCS, C315GS-4). The tip of the canula was inserted below the dura at the very edge of the craniotomy. The canula and cranial window were then cemented in place together[59–61].

**Image acquisition.** Images were acquired using an Andor iXon Ultra 888 EMCCD camera and a ×16 0.8 NA objective (Nikon CFI75) using Andor Solis data acquisition software (Andor Solis 64 bit, v.4.32). Imaging data of the CaBLAM mice were acquired under no illumination in a light shielded enclosure (512 × 512 pixels after 2 × 2 binning) at a frame rate of 10 Hz (0.0955 s exposure) and an electron multiplication gain of 300. The GCaMP6s mice were imaged under epi-illumination from a X-Cite 120Q mercury vapor lamp (Excelitas Technologies) using an enhanced green fluorescent protein (EGFP) filter set (Chroma 49002, excitation filter: ET470/×40, dichroic: T495lpxr, emission filter: ET525/50 m). The imaging parameters were otherwise the same as in the CaBLAM mice with the exception that the electron multiplication gain was set to 0.

The pan-neuronal CaBLAM mice were imaged using the same settings as above, with the exceptions of the use of a ×4 0.13 NA objective (Olympus 1-U2B5222) at rates of 2 Hz and 10 Hz (0.495 and 0.0955 s exposures). One of these mice was also imaged for an extended period (~5 h) under anesthesia after direct application of FFz. In this experiment images were acquired at 10 Hz as above, with all other parameters unchanged with the exception of the use of a ×10 0.30 NA objective (Olympus 1-U2B5242). At the end of the 5-h imaging session, to determine whether BL subcellular neural processes can be identified using CaBLAM, we imaged at 1 Hz (no binning, 300 electron multiplication gain) using a ×40 0.8 NA objective (Olympus 1-U2M587). Image stacks were acquired from multiple ROIs for ~1 min each.

**Experimental procedures.** Experiments were conducted 3–8 weeks postsurgery. For all animals, throughout data collection, a tactile stimulus was delivered to the right mystacial vibrissa pad using a piezo bender (Noliac). Tactile stimuli consisted of asymmetric sinusoidal deflections (10 ms rise and 15 ms fall time, 5 repetitions, 125 ms total stimulus length) delivered at random intervals between 20 s and 25 s. Each animal received a minimum of 20 stimulus presentations. Animals were head-fixed and allowed to freely run on a wheel.

During the direct cortical infusions of luciferin, the dummy cannula was carefully removed and replaced with an infusion cannula (Plastics One, C315IS-4) attached to a length of tubing connected to a 5-µl syringe (Hamilton 87930). FFz was reconstituted in sterile water (8.76 mM; 4.6 µmol per 525 µl) and infused at a rate of 50–200 nl min⁻¹ using a motorized injector (WPI UMP3) for a total volume of 500 nl. The infusion cannulas were kept in place for the duration of the imaging session.

For peripheral luciferin delivery, CFz9 (Promega) was reconstituted in 0.2 M Tris buffer (8.40 mM; 8.40 µmol per 1 ml) and 200 µl of the solution was injected retro-orbitally immediately before (<1 min) the start of imaging.

To assess the viability of long-duration imaging using CaBLAM, we removed the cortical window from one of the pan-neuronal CaBLAM mice under anesthesia (1–2% isoflurane). The mouse remained under anesthesia for the duration of the experiment that lasted ~5 h. Next, 50 µl of FFz, reconstituted as above, was pipetted into a saline well over the craniotomy. Tactile stimuli we administered as above throughout the recording. The health and anesthetic depth of the mouse was monitored by breathing rate and toe pinch, checked at intervals of ~20 min, during which the recording was paused.

Blinded data acquisition was not feasible for these experiments due to the substantial differences in imaging procedures between GCaMP6s and CaBLAM.

**Data analysis.** Offline analyses of the in vivo imaging data from both CaBLAM and GCaMP6s were performed identically in Python v.3.9.22 and MATLAB v.R2024b (MathWorks) using custom scripts as well as the MATLAB toolbox SUPPORT (https://github.com/NICALab/SUPPORT, ref. [62]) and the Python package Suite2p (https://github.com/MouseLand/suite2p, v.0.14.0, ref. [63]). Investigators were blinded to the identity of the sensor during analysis. The raw images were first spatially and temporally denoised using the SUPPORT toolbox using a pretrained model available on the authors' GitHub (bs3.pth). Using Suite2p, the denoised data was then motion corrected, automatically segmented into ROIs (followed by manual curation), and neuropil masks were created. The raw ROI traces were then imported into MATLAB. The neuropil masks were used to apply a neuropil correction,

$$X(t) = X_{raw}(t) - r \times X_{neuropil}(t)$$

where $X_{raw}$ is the fluorescent or BL signal of the ROI, $X_{neuropil}$ is the signal from the surrounding neuropil mask and $r$ is the decontamination factor, which was set to 1 (ref. [64]). The resultant time series were smoothed using a 5-point moving average window.

To analyze stimulus-evoked activity, we extracted −3 to 7 s windows centered on stimulus onset for each stimulus presentation and all ROIs. The prestimulus baseline period −3 to 0 s before stimulus onset was then used to calculate $\Delta F/F_0$ or $\Delta L/L_0$ for the fluorescent and BL data, respectively. We then identified responsive cells as those whose bootstrapped 95% CIs across all stimulus presentations in at least one of two response windows (0–1 s and 1–2 s) fell outside those of the cross-trial bootstrapped 95% CI of the baseline period. This yielded both positively and negatively responsive cells relative to baseline[30]. No data were excluded from this analysis. Analyses included 7 CaBLAM and GCaMP6s mice with 82 CaBLAM-responsive and 44 GCaMP6s-responsive cells contributing to latency analyses and >3,100 single-trial SNR measurements (Fig. 4). Sample sizes followed precedent from previous NDNF-Cre imaging studies and were sufficient to achieve highly significant differences in latency and SNR.

### In vivo Ca²⁺ imaging in zebrafish
**Fish husbandry and sample preparation.** All procedures involving larval zebrafish (*Danio rerio*) were approved by the Institutional Animal Care and Use Committee at New York University Grossman School of

Medicine. Adult zebrafish were maintained at 28.5 °C under a standard 14/10-h light/dark cycle. Embryos were raised at densities ranging from 20 to 50 in 10 cm diameter petri dishes, each containing 25 to 40 ml of E3 medium with 0.5 ppm methylene blue added. At 1 dpf, larvae were kept in E3 medium without methylene blue. Larvae were screened for expression using a fluorescent stereomicroscope (Leica M165FC). Larvae were soaked in an E3 solution containing either 1:100 or 1:1,000 vivazine (Promega Nano-Glo N2580) for 10 min and then mounted in 2% low melting point agarose. The tail (posterior to the pectoral fins) was then freed, and larvae remained in vivazine for the duration of the experiment.

**Zebrafish lines.** All larvae used were on the *mitfa*$^{-/-}$ background to remove pigment cells. Existing driver lines were *Tg(-6.7Tru.Hcrtr2: GAL4-VP16)*[35], called *Tg(hcrtr2)*, stl601Tg[34], called *Is(nefma)* and psi1Tg[36], called *Tg(elavl3)*. In addition, two new transgenic lines were generated for this study.

First, the expression construct pTol2_slc1a3b:KalTA4 was generated using Gateway cloning to recombine p5E_slc1a3b[33], pME_KalTA4 (ref. [65]) and p3E_pA with pDestTol2CG2 from the Tol2Kit[66]. The resulting construct was microinjected into fertilized zebrafish eggs (using 1 nl of an injection solution containing 20 ng µl$^{-1}$ DNA, 50 ng µl$^{-1}$ Tol2 transposase messenger RNA and 10% phenol red). Injected F0 animals were outcrossed to wild-types and F1 offspring was screened for germline transmission to obtain *Tg(slc1a3b:KalTA4)*, called *Tg(GLAST)*.

Second, the mNeonGreen-CaBLAM sequence was optimized for zebrafish translation using CodonZ[67]. The codon-optimized sequence was then synthesized (VectorBuilder) into a zebrafish Tol2 expression vector following 5xUAS before a SV40 PolyA tail. The expression plasmid was injected into *Is(nefma)* embryos at the one-cell stage, screened for green fluorescence, and raised to adulthood. Injected fish were outcrossed to wild-type fish to screen for founders. F2 and F3 embryos were used for experiments. All fish used for experiments were monoallelic for both the driver and CaBLAM.

**Imaging and analysis.** Anatomical imaging was done on a confocal microscope (Zeiss LSM 800) with a ×20/1.0 NA water-dipping objective. Acquisitions were tiled to cover the entire length of the fish.

Simultaneous measurements of bioluminescence and behavior were made with a custom microscope that consisted of a high-sensitivity machine vision camera (Ximea MC023MG-SY with a SONY IMX174 sensor), a zoom lens (Navitar Zoom 7000, ×1 at f/5.6), a custom chamber made from a magnetic mount (ThorLabs CP44F) and a glass-bottom dish (WPI FluoroDish FD3510), illuminated by a strip of 850-nm LEDs. Flux was measured with a photon-counting PMT (Hamamatsu H11890-110, 8 mm window) behind a 25-mm bandpass filter (ThorLabs BG-39) to block infrared light. The entire microscope was enclosed in a light-tight double-walled enclosure; when the infrared light strip was off the PMT was at its noise floor (100 counts per s after 15 min). The microscope was controlled using custom software (LabView 2021) to acquire images (600 × 600 pixels, 250 fps, 1 ms or 2 ms exposure time) and sample from the PMT (40 Hz). Movement amplitude was defined as the number of pixels that changed intensity over a noise threshold (15/255) from frame to frame.

All analyses took place using custom code written in MATLAB (v.2024b, MathWorks). To identify high-amplitude movements, we first defined a threshold that would reliably identify the largest events. To account for differences in imaging settings and variation in IR-reflectivity of each fish, we set the threshold at either the 99.9th percentile (1 ms exposure time) or 99.9975th percentile (2 ms exposure time) of movements for each fish. The first threshold-crossing event in a given second was defined to be the beginning of each high-amplitude, and events were spot-checked to ensure that they were comparable (that is prolonged and uncoordinated large-amplitude tail movements) by post hoc examination of saved videos. We processed the vector of

counts from the PMT by interpolating with a spline fitting algorithm to match the timebase of behavior and then smoothing the vector with a 0.25 s square window. For each fish, we extracted a response defined as the counts 2 s before and 30 s after a threshold-crossing event. The baseline was defined as the mean of the first 2 s of the response. To facilitate comparison across genotypes, we subtract the baseline from the response and divide the result by the baseline that is (counts($t$) − baseline)/baseline. No data were excluded from this analysis. Zebrafish imaging included three head-fixed larvae per genotype and their nonfluorescent siblings where applicable (Fig. 5h–k and Extended Data Table 1). No statistical power analysis was conducted; the number of individuals per genotype was selected to confirm consistency of movement-coupled BL responses across independent larvae.

Widefield imaging of bioluminescence was performed in two ways. First, slow images (30 s exposures) were taken with the machine vision camera above with the aperture set at f/2.8. Images were binned 4 × 4, and a single image was generated by taking the s.d. across pixels in the resulting stack (83 frames), and then up-sampling 4 × 4 to overlay with an infrared-illuminated reference image. Second, high-speed imaging was performed using an intensified camera (HiCAM Fluo, Lambert Instruments) through with a Cousa ×10/0.5 NA air objective[68]. A sequence of images was captured over 2 min at 20 frames per second. Raw images were processed by subtracting a dark-count background image, smoothed with (1) a 3-frame rolling window, (2) a 2 × 2 Gaussian blur and (3) a 3-pixel median filter in ($x,y,t$). A single-frame average projection was then generated to represent basal intensity. The final result was generated by subtracting the basal intensity from each image slice. No data were excluded from this analysis.

### Reporting summary

Further information on research design is available in the Nature Portfolio Reporting Summary linked to this article.

### Data availability

Raw and processed datasets from experiments performed in this study are freely available via the Brown Digital Repository[69–71] at https://doi.org/10.26300/7sg5-w257 in vivo, https://doi.org/10.26300/fcw9-qp80 in vitro and https://doi.org/10.26300/b2df-k359, and as input data for reproducing analysis with the custom code developed in this study available via GitHub at https://github.com/Shaner-Lab/CaBLAM. Coding sequences for CaBLAM and GeNL_SS have been deposited to GenBank (accession nos. PV987411 and PV987412). Mammalian expression plasmids (pCAG) encoding GeNL_SS, CaBLAM, CaBLAM_294W and CaBLAM_332W and AAV plasmids encoding CaBLAM driven by hSyn, GFAP and EF1a-DIO are deposited with AddGene (plasmid numbers 244128, 244129, 244130, 244131, 244227, 244228 and 244229). All other unique materials are available by request to the corresponding author.

### Code availability

Custom code used in data collection, processing and analysis is freely accessible via GitHub at https://github.com/Shaner-Lab/CaBLAM under the CC0 1.0 Universal license.

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

## Acknowledgements

We thank several individuals who supported this project as allies: S. Adams for many years of insightful discussions about sensors, Ca²⁺ buffers and other subjects; P. Wales for her invaluable help with camera and objective selection and optimization; K. Monk (Vollum Institute, Oregon Health & Science University) for providing zebrafish lines; D. Black (University of California, Los Angeles) for providing Neuro 2a cells; M. Lovett-Barron for helpful advice on zebrafish experiments; and L. Flores and M. Oberholzer for invaluable scientific discussions and moral support through very challenging times. We also thank all members of our Bioluminescence Hub (http://www.bioluminescencehub.org/) laboratories for their feedback, discussions and thoughtful comments throughout the progression of this work. This work was supported by the National Institutes of Health (grant nos. R21EY030716 (N.C.S.), R21MH101525 (U.H.), R01GM121944 (N.C.S.), R01CA279813 (N.C.S.), R01NS120832 (C.I.M., U.H. and N.C.S.), R21EY036659 (N.C.S.), R34DA059500 (K.I.N., D.S. and N.C.S.), R21NS115437 (J.C.), R01EY035691 (D.S.), R01MH124811 (A.A.), F32NS134617 (K.L.T.), U01NS099709 (C.I.M., U.H. and N.C.S.) and F99NS129170 (E.L.C.)), the National Science Foundation (CBET-1464686 (U.H. and N.C.S.), NeuroNex DBI-1707352 (D.L., C.I.M., U.H. and N.C.S.) and DBI-2208914 (B.N.S.)), a Research Seed Award from Brown University's Office of the Vice President for Research (A.A.) and the Allen Discovery Center for Neurobiology in Changing Environments, funded by the Paul G. Allen Frontiers Group (N.C.S.). The funders had no role in study design, data collection and analysis, decision to publish or preparation of the paper.

## Author contributions

Conceptualization: G.G.L., E.L.C., D.L., C.I.M., U.H. and N.C.S. Validation: G.G.L., E.L.C., J.M., S.V., D.C., D.L., C.I.M., U.H. and N.C.S. Formal analysis: G.G.L., E.L.C., J.M., D.S. and N.C.S. Investigation: K.L.T., D.B., J.H., S.L., D.K.N., B.N.S., M.O.T., S.V., D.C., R.O., D.H., A.B.B., A.T.Z., A.H., J.C., R.M., H.G., E.G., Y.Z., M.K., D.S. and N.C.S. Data curation: G.G.L., E.L.C., J.M., K.I.N., D.S., S.S., Y.Z. and N.C.S. Writing—original draft: G.G.L., E.L.C. and N.C.S. Writing—review, revision and editing: G.G.L., E.L.C., J.M., K.L.T., M.C., B.N.S., D.L., C.I.M., U.H., J.J.A., A.A., Y.Z., S.S., D.S. and N.C.S. Visualization: G.G.L., E.L.C., J.M., D.S. and N.C.S. Supervision: D.L., A.A., C.I.M., U.H., T.C., S.S., D.S., K.I.N. and N.C.S. Project administration: U.H., J.J.A., T.C., K.I.N., D.S. and N.C.S. Funding acquisition: D.L., C.I.M., U.H., A.A., B.N.S., J.C., K.I.N., D.S. and N.C.S.

## Competing interests

The authors declare no competing interests.

## Additional information

**Extended data** is available for this paper at https://doi.org/10.1038/s41592-025-02972-0.

**Correspondence and requests for materials** should be addressed to Nathan C. Shaner.

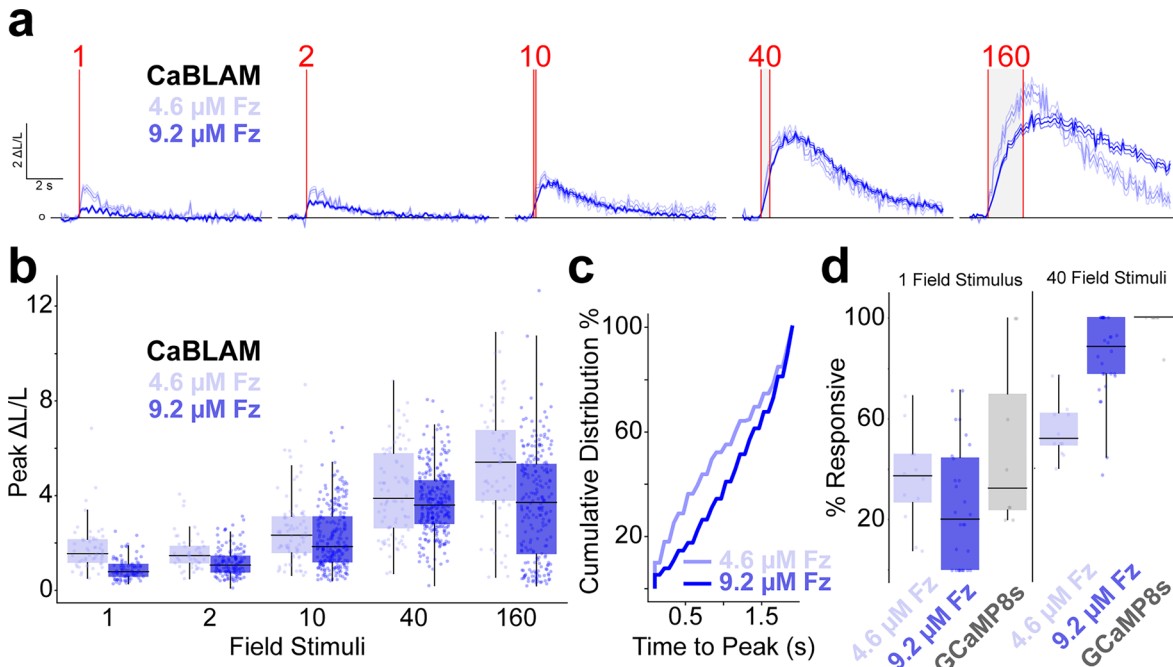

**Extended Data Fig. 1 | Characterizing furimazine dose-dependent performance of CaBLAM in cultured neurons. a**, Overlaid bioluminescence ΔL/L time-locked traces of CaBLAM Ca²⁺ responses to 1, 2, 10, 40, 160 pulses of 1 ms field stimulations at 83 Hz, at 4.6 μM or 9.2 μM Fz concentration. Red dashed lines indicate field stimulation window. Data are shown as mean ± s.e.m. **b**, Peak stimulus ΔL/L for CaBLAM across neurons elicited across increasing field stimulations. **c**, Cumulative distribution of time to ΔL/L responses between (includes statistical analysis). **d**, Proportion of neurons responding to 1 (left) and 40 (right) electrical field stimulations. All boxplots show the median, 25th and 75th percentiles (box edges), whiskers extending to the most extreme data points, and individual outliers plotted separately.

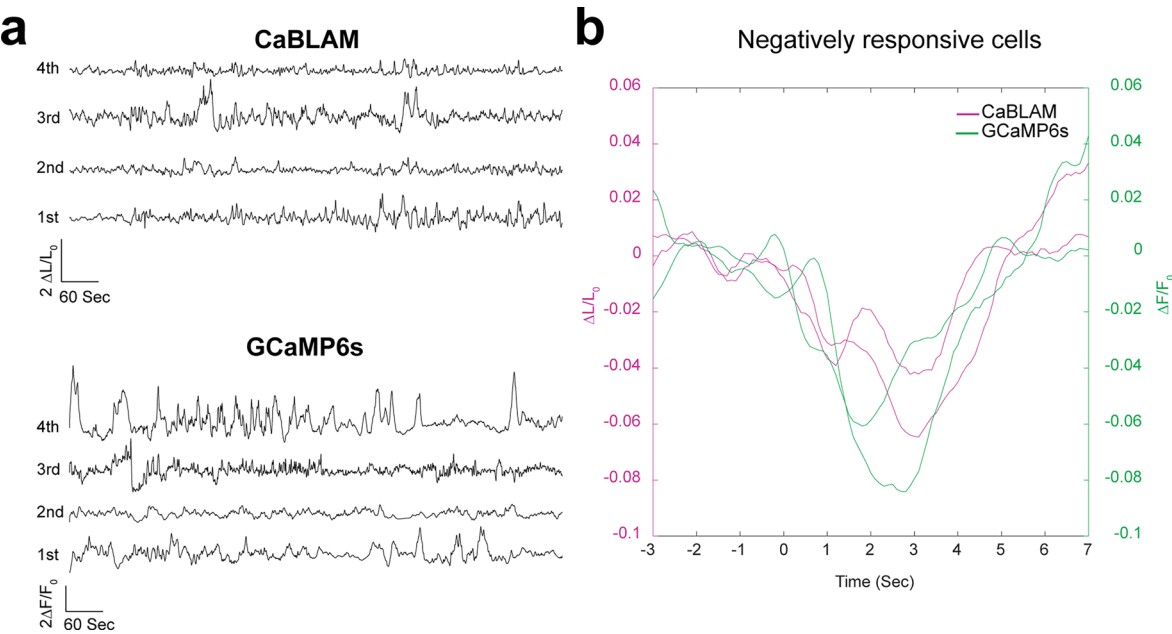

**Extended Data Fig. 2 | Classifying cell responses to vibrissa stimulation. a**, Example 8-min CaBLAM and GCaMP6s traces. In both, positively responsive cells were sorted by SNR. From bottom to top, traces from the 1st, 2nd, 3rd and 4th SNR quartiles **b**, Example of average response from 2 negatively responsive cells in the CaBLAM and GCaMP6s groups. Note the left and right y-axes.

**a**

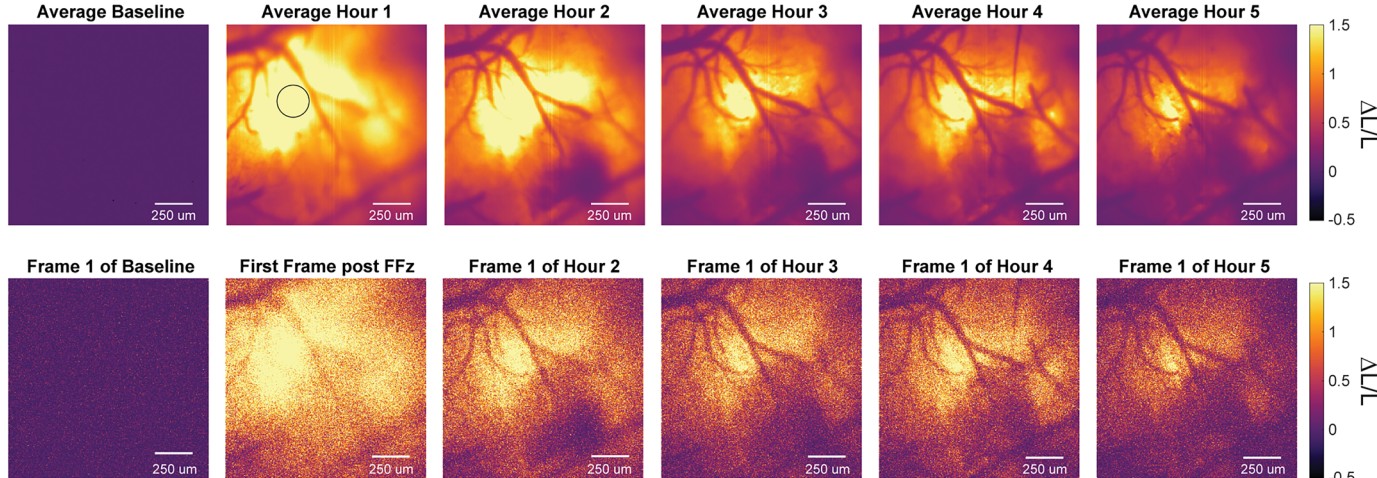

**b**

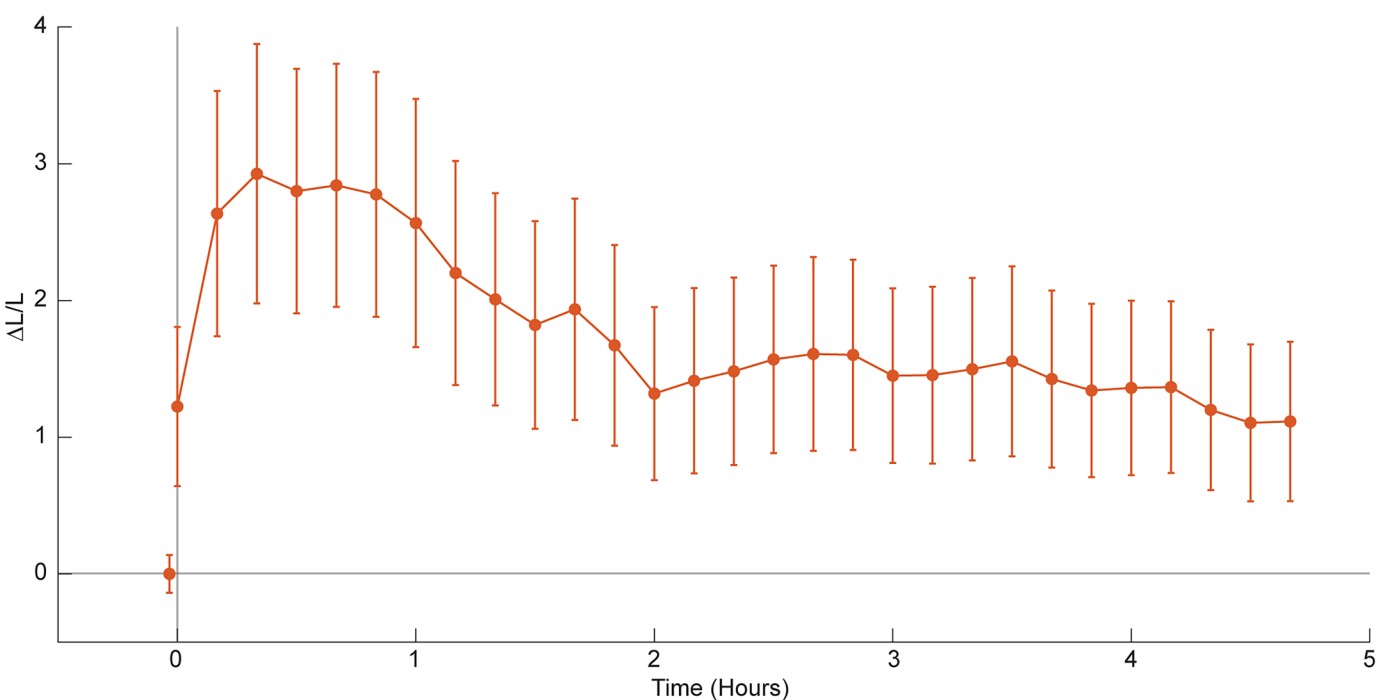

**Extended Data Fig. 3 | Bioluminescence time course with a single direct cortical application of FFz in a pan-neuronal CaBLAM mouse.** All data is expressed in units of $\Delta L/L_0$, where baseline ($L_0$) consisted of a 1 min recording prior to FFz administration (first orange marker in **b**). The imaging was then paused for ~2 min in order to administer the FFz, after which recording commenced (time 0). **a**, *top row*, average projection images from each hour (600 images/h); *bottom row*, first single frames from each hour after FFz administration. **b**, Bioluminescence from a circular ROI (black circle in **a**, top row, second from left) binned at 10-min intervals. Error bars represent ± 1 SD across all pixels in the ROI.

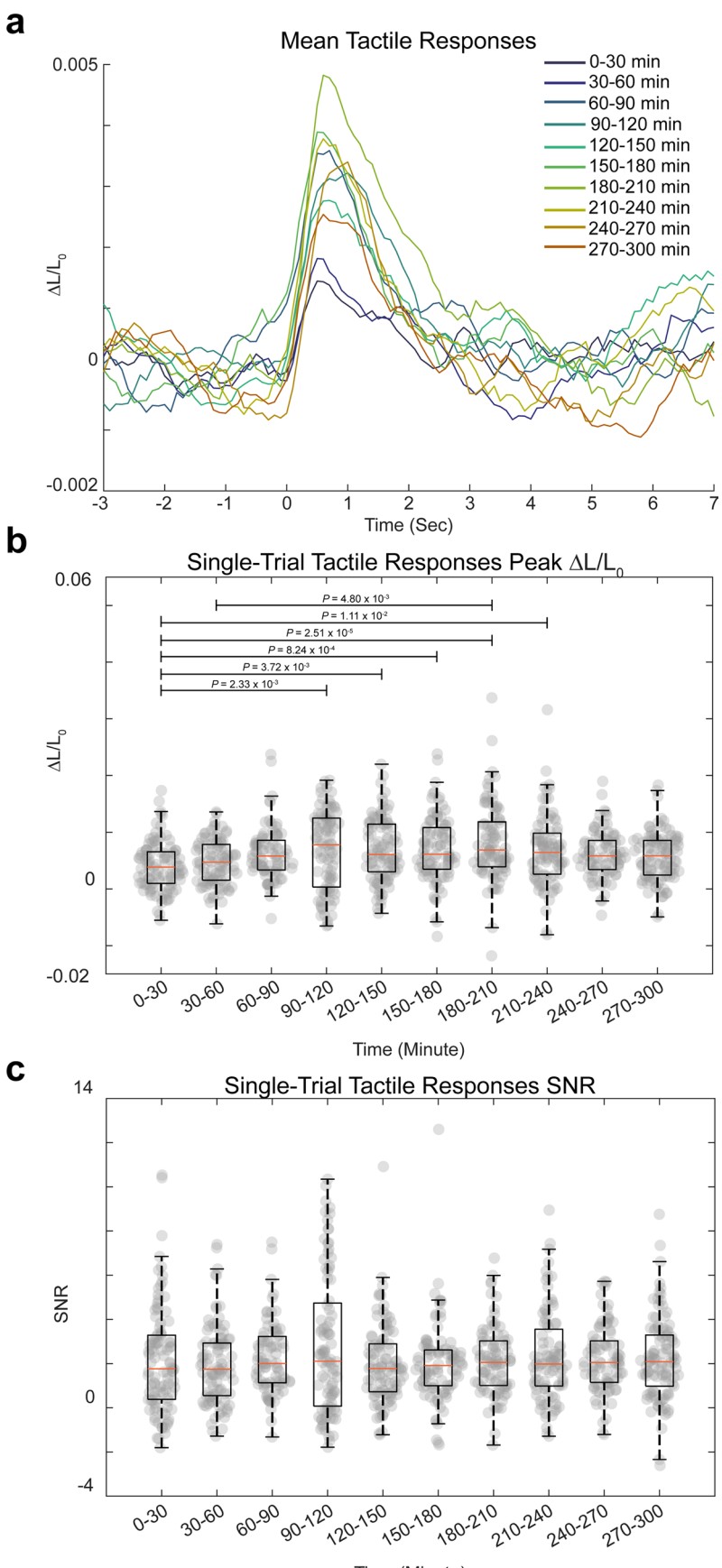

**Extended Data Fig. 4 | See next page for caption.**

**Extended Data Fig. 4 | Tactile responses over time in an anaesthetized pan-neuronal CaBLAM mouse with a single direct cortical application of FFz.**
**a**, Average CaBLAM tactile response from ROI marked in Extended Data Fig. 3a, binned at 30-min intervals after the administration of FFz. **b**, Box plot of the single-trial peak $\Delta L/L_0$ binned at 30-min intervals. Orange lines indicate the median, boxes enclose the middle 2 quartiles of the data, whiskers extend 1.5 IQRs above and below. Gray dots are single data points. A Kruskal-Wallace test indicated a significant interaction of peak $\Delta L/L_0$ single-trial responses and time bin (10 30-min bins; $P = 4.52 \times 10^{-6}$; $\chi^2 = 41.24$, df = 9). $P$-values from follow-up comparisons (Tukey's HSD) are indicated above horizontal black bars. **c**, box plot of SNR, same conventions as in **b**. SNR was not significantly different across time bins (Kruskal-Wallace test, $P = 0.82$, $\chi^2 = 5.11$, df = 9).

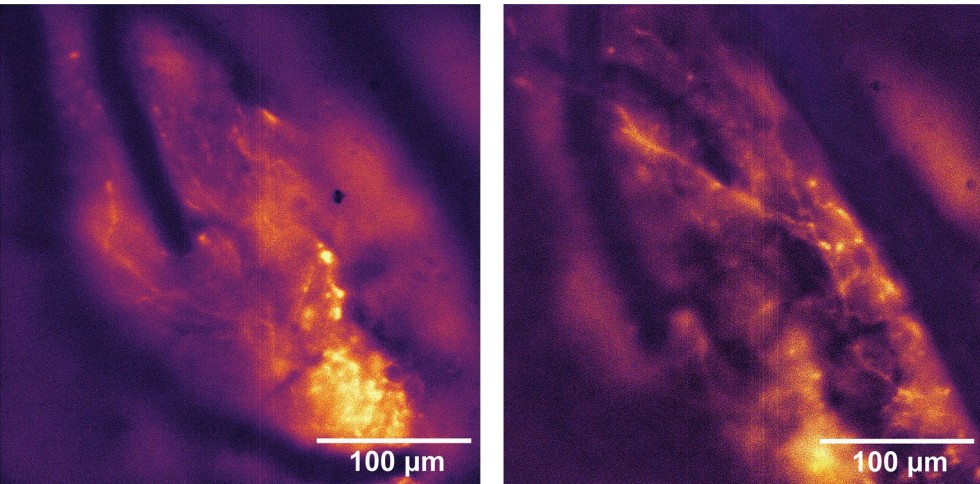

**Extended Data Fig. 5 | Two images of neural processes acquired at 40x *after* the 5-h recording from one pan-neuronal CaBLAM mouse.** Images were acquired at 1 Hz with no binning at 300 EM gain. Displayed images are averaged from stacks of 32 (*left*) and 31 (*right*) frames, with pixel values scaled to the range of the data for each.

**Extended Data Table 1 | Parameters for zebrafish CaBLAM experiments**

| Genotype | Vivazine concentration | Average bioluminescence (counts / s)* | Number of high-amplitude events* | Experiment duration (min)* |
|---|---|---|---|---|
| Is(nefma) | 1:100 | 16631<br>17818<br>13793 | 25<br>10<br>12 | 81<br>37<br>41 |
| Is(nefma) sibling controls | 1:100 | 3007<br>2393<br>2692 | 11<br>15<br>8 | 39<br>60<br>34 |
| Tg(GLAST) | 1:1000 | 113664<br>20347<br>15280 | 10<br>8<br>6 | 32<br>28<br>28 |
| Tg(hcrtr2) | 1:100 | 16525<br>22896<br>32134 | 16<br>95<br>38 | 29<br>103<br>45 |
| Tg(elavl3) | 1:1000 | 36712<br>87159<br>67205 | 57<br>22<br>5 | 125<br>41<br>57 |

\* Values shown are from three experimental runs with individual zebrafish larvae for each genotype.

# Reporting Summary

## Statistics

For all statistical analyses, confirm that the following items are present in the figure legend, table legend, main text, or Methods section.

| n/a | Confirmed | |
|---|---|---|
| ☐ | ☒ | The exact sample size (*n*) for each experimental group/condition, given as a discrete number and unit of measurement |
| ☐ | ☒ | A statement on whether measurements were taken from distinct samples or whether the same sample was measured repeatedly |
| ☐ | ☒ | The statistical test(s) used AND whether they are one- or two-sided<br>*Only common tests should be described solely by name; describe more complex techniques in the Methods section.* |
| ☒ | ☐ | A description of all covariates tested |
| ☐ | ☒ | A description of any assumptions or corrections, such as tests of normality and adjustment for multiple comparisons |
| ☐ | ☒ | A full description of the statistical parameters including central tendency (e.g. means) or other basic estimates (e.g. regression coefficient) AND variation (e.g. standard deviation) or associated estimates of uncertainty (e.g. confidence intervals) |
| ☒ | ☐ | For null hypothesis testing, the test statistic (e.g. *F*, *t*, *r*) with confidence intervals, effect sizes, degrees of freedom and *P* value noted<br>*Give P values as exact values whenever suitable.* |
| ☒ | ☐ | For Bayesian analysis, information on the choice of priors and Markov chain Monte Carlo settings |
| ☒ | ☐ | For hierarchical and complex designs, identification of the appropriate level for tests and full reporting of outcomes |
| ☒ | ☐ | Estimates of effect sizes (e.g. Cohen's *d*, Pearson's *r*), indicating how they were calculated |

*Our web collection on statistics for biologists contains articles on many of the points above.*

## Software and code

Policy information about availability of computer code

Data collection | Data acquisition used a combination of commercial and open-source software: NIS-Elements AR v5.41.01 (Nikon Instruments) for microscopy control and image capture, CLARIOstar v5.40 E4 (BMG LABTECH) for plate reader measurements, Andor Solis 64-bit v4.32.30000.0 (Oxford Instruments) for EMCCD data acquisition, Arduino IDE v2.3.2 for device control, and Bonsai v2.8.5 for synchronized behavioral and photometric recordings.

Data analysis | Data were analyzed using Fiji/ImageJ v1.54d–f (NIH), Python v3.9.22, and MATLAB R2024b, with additional open-source packages including SUPPORT (https://github.com/NICALab/SUPPORT, accessed June 4 2025) and Suite2p v0.14.0 (https://github.com/MouseLand/suite2p). Custom Python and MATLAB analysis scripts are available at https://github.com/Shaner-Lab/CaBLAM.

For manuscripts utilizing custom algorithms or software that are central to the research but not yet described in published literature, software must be made available to editors and reviewers. We strongly encourage code deposition in a community repository (e.g. GitHub). See the Nature Portfolio guidelines for submitting code & software for further information.

# Data

Policy information about availability of data

All manuscripts must include a data availability statement. This statement should provide the following information, where applicable:
- Accession codes, unique identifiers, or web links for publicly available datasets
- A description of any restrictions on data availability
- For clinical datasets or third party data, please ensure that the statement adheres to our policy

Raw and processed data sets from experiments performed in this study are freely available via the Brown Digital Repository (https://doi.org/10.26300/7sg5-w257) and as input data for reproducing analysis with the custom code developed in this study (https://github.com/Shaner-Lab/CaBLAM). Bacterial and mammalian expression plasmids encoding SSLuc, GeNL_SS, and CaBLAM will be deposited for distribution by AddGene (pending); prior to availability from AddGene, plasmids will be shared with non-profit researchers upon request to NCS.

# Human research participants

Policy information about studies involving human research participants and Sex and Gender in Research.

| | |
|---|---|
| Reporting on sex and gender | N/A |
| Population characteristics | N/A |
| Recruitment | N/A |
| Ethics oversight | N/A |

Note that full information on the approval of the study protocol must also be provided in the manuscript.

# Field-specific reporting

Please select the one below that is the best fit for your research. If you are not sure, read the appropriate sections before making your selection.

☒ Life sciences  ☐ Behavioural & social sciences  ☐ Ecological, evolutionary & environmental sciences

For a reference copy of the document with all sections, see nature.com/documents/nr-reporting-summary-flat.pdf

# Life sciences study design

All studies must disclose on these points even when the disclosure is negative.

| | |
|---|---|
| Sample size | No formal power analyses were conducted. Sample sizes were chosen based on prior experience with similar imaging and biochemical experiments to ensure sufficient replication for reproducible estimates of variance and statistical testing. Replicate numbers matched established practice for genetically encoded indicator characterization, and observed effect sizes consistently exceeded within-group variability. |
| Data exclusions | No data were excluded from any analysis. |
| Replication | All key experiments were independently repeated with separate biological or technical replicates and reproduced similar results. In vitro, cell-based, and in vivo datasets were validated across multiple preparations or animals, and all replication attempts yielded consistent outcomes within expected biological variation. |
| Randomization | NDNF-Cre individual mice were assigned randomly to receive AAV encoding CaBLAM or GCaMP6s. |
| Blinding | For in vivo comparisons of CaBLAM and GCaMP6s, Investigators were blinded to the identity of the sensor during analysis. Blinding was not practical for other experiments or analysis. |

# Reporting for specific materials, systems and methods

We require information from authors about some types of materials, experimental systems and methods used in many studies. Here, indicate whether each material, system or method listed is relevant to your study. If you are not sure if a list item applies to your research, read the appropriate section before selecting a response.

## Materials & experimental systems

| n/a | Involved in the study |
|---|---|
| ☒ | Antibodies |
| ☐ | ☒ Eukaryotic cell lines |
| ☒ | Palaeontology and archaeology |
| ☐ | ☒ Animals and other organisms |
| ☒ | Clinical data |
| ☒ | Dual use research of concern |

## Methods

| n/a | Involved in the study |
|---|---|
| ☒ | ChIP-seq |
| ☒ | Flow cytometry |
| ☒ | MRI-based neuroimaging |

# Eukaryotic cell lines

Policy information about cell lines and Sex and Gender in Research

| | |
|---|---|
| Cell line source(s) | HeLa and U2-OS cell lines were obtained from ATCC. N2a cells were a gift from Douglas Black. |
| Authentication | Cell lines were not authenticated in this study, but HeLa and U2-OS were grown directly from ATCC stocks, which have been validated by ATCC. |
| Mycoplasma contamination | All cell lines tested negative for mycoplasma. |
| Commonly misidentified lines (See ICLAC register) | No commonly misidentified cell lines were used in this study. |

# Animals and other research organisms

Policy information about studies involving animals; ARRIVE guidelines recommended for reporting animal research, and Sex and Gender in Research

| | |
|---|---|
| Laboratory animals | Cortices were dissected out from one P2 Sprague-Dawley rat for neuron culture. Rat pups were housed with the dam in a vivarium on a 12-hour reversed light–dark cycle and had free access to food and water.<br><br>Seven NDNF-Cre mice (2 female/5 male; 23-35 weeks old on imaging day; JAX stock #030757) were used for in vivo imaging to selectively express CaBLAM or GCaMP6s in neuron-derived neurotrophic factor (NDNF) expressing cortical layer 1 interneurons. Three additional NDNF-Cre mice (0 female/3 male, each 14 weeks old on imaging day) were injected with a pan-neuronal CaBLAM to test peripheral luciferin delivery. Mice were housed in a vivarium on a reversed light-dark cycle and had free access to food and water.<br><br>15 larval zebrafish (Danio rerio), 1 day post fertilization, were used for imaging experiments, including 3 individuals from each of the following transgenic lines: Is(nefma), Is(nefma) (non-transgenic sibling controls), Tg(GLAST), Tg(hcrtr2), and Tg(elavl3). All larvae used were on the mitfa -/- background to remove pigment cells. Existing driver lines were Tg(-6.7Tru.Hcrtr2:GAL4-VP16) 19, called Tg(hcrtr2), stl601Tg20, called Is(nefma), and psi1Tg21, called Tg(elavl3). In addition, two new transgenic lines were generated for this study. Adult zebrafish were maintained at 28.5 ºC under a standard 14/10-hour light/dark cycle. |
| Wild animals | N/A |
| Reporting on sex | Aside from using both male and female mice, sex was not considered in the study design. Zebrafish sex was not determined for the individuals imaged in this study. Sex was not determined for the rat pup used to prepare cortical neurons. |
| Field-collected samples | N/A |
| Ethics oversight | All procedures involving rats were approved by the Institutional Animal Care and Use Committee of UC San Diego.<br><br>All procedures involving mice were conducted in accordance with the guidelines of the National Institute of Health and with approval of the Animal Care and Use Committee of Brown University.<br><br>All procedures involving larval zebrafish (Danio rerio) were approved by the Institutional Animal Care and Use Committee (IACUC) at New York University Grossman School of Medicine. |

Note that full information on the approval of the study protocol must also be provided in the manuscript.

