## [Peer Review File · Nature Methods]

CaBLAM: A high-contrast bioluminescent Ca²⁺ indicator derived from an engineered *Oplophorus gracilirostris* luciferase

Corresponding Author: Dr Nathan Shaner

Version 0:

Decision Letter:

21st Nov 2024

Dear Nathan,

Thank you for your inquiry about submitting your manuscript, "CaBLAM! A high-contrast bioluminescent Ca²⁺ indicator derived from an engineered *Oplophorus gracilirostris* luciferase" to Nature Methods. The paper sounds interesting, and should fit the scope of the journal. We would be pleased to consider it further for publication in Nature Methods.

Our presubmission inquiry function is intended to give rapid feedback to authors about the potential suitability of a manuscript for Nature Methods. Therefore, please note that it is our policy to not read papers in full at the presubmission inquiry stage; we base our recommendations mainly on the cover letter and abstract. Therefore, I am sure you will understand that we cannot make any specific promises, even about sending a paper out for peer review, until we have read it in its entirety.

Just as a note, my colleague Dr. Nina Vogt will likely be the primary editor handling your manuscript upon full submission, as she handles most of our neuroscience content including sensors. I have cc'd her on this email in case you have any questions. Either way, we will both discuss the full submission.

Please keep in mind that the journal is aimed at a large, interdisciplinary audience and places a strong emphasis on the practical value of the work presented for basic research in the life sciences. We strongly encourage you to include data to validate method performance and demonstrate its general applicability.

You will find our Guide to Authors at <http://www.nature.com/naturemethods> to assist you in preparing your manuscript. However, it is not necessary at this stage to spend major effort adhering to our detailed formatting instructions.

Again, thank you for your interest in Nature Methods, and we look forward to reading your manuscript.

Sincerely,
Rita

Rita Strack, Ph.D.
Senior Editor
Nature Methods

Version 1:

Decision Letter:

21st Feb 2025

Dear Dr Shaner,

Your Article, "CaBLAM! A high-contrast bioluminescent Ca²⁺ indicator derived from an engineered *Oplophorus gracilirostris* luciferase", has now been seen by three reviewers. As you will see from their comments below, although the reviewers find your work of considerable potential interest, they have raised a number of concerns. We are interested in the possibility of publishing your paper in Nature Methods, but would like to consider your response to these concerns before we reach a final decision on publication.

We therefore invite you to revise your manuscript to address these concerns. Specifically, I would like to highlight the need for additional in vivo demonstrations showcasing the sensor's utility. Please also keep in mind that the current comparison of different sensors ion awake and anesthetized animals is not ideal. And I encourage you to show that the CaBLAM signals can be analyzed with automated tools.

Link Redacted

We hope to receive your revised paper within 2-3 months. If you cannot send it within this time, please let us know. In this event, we will still be happy to reconsider your paper at a later date so long as nothing similar has been accepted for publication at Nature Methods or published elsewhere.

OPEN SCIENCE REQUIREMENTS

REPORTING SUMMARY AND EDITORIAL POLICY CHECKLISTS

IMAGE INTEGRITY

When submitting the revised version of your manuscript, please pay close attention to our >Digital Image Integrity Guidelines and to the following points below:

EXTENDED DATA FIGURES

DATA AVAILABILITY

All novel DNA and RNA sequencing data, protein sequences, genetic polymorphisms, linked genotype and phenotype data, gene expression data, macromolecular structures, and proteomics data must be deposited in a publicly accessible database, and accession codes and associated hyperlinks must be provided in the "Data Availability" section.

MATERIALS AVAILABILITY

ORCID

Nature Methods is committed to improving transparency in authorship. As part of our efforts in this direction, we are now requesting that all authors identified as 'corresponding author' on published papers create and link their Open Researcher and Contributor Identifier (ORCID) with their account on the Manuscript Tracking System (MTS), prior to acceptance. This applies to primary research papers only. ORCID helps the scientific community achieve unambiguous attribution of all scholarly contributions. You can create and link your ORCID from the home page of the MTS by clicking on 'Modify my Springer Nature account'. For more information please visit please visit <http://www.springernature.com/orcid>.

Best regards,
Nina

Nina Vogt, PhD
Senior Editor
Nature Methods

Reviewers' Comments:

Reviewer #1 (Remarks to the Author):

Lambert et al. reported the development of a novel bioluminescent calcium indicator, "CaBLAM," based on SSLuc, an evolved variant of OLuc. This indicator offers significantly higher contrast for detecting physiologically relevant calcium dynamics in cells and in vivo compared to previous bioluminescent indicators. The authors characterized the performance of CaBLAM, benchmarking it against GeNL(Ca²⁺)₄₈₀ in mammalian cells and dissociated neurons, as well as against the recently optimized GCaMP8s in dissociated neurons and the cortex in vivo. CaBLAM represents a major advancement in bioluminescent calcium indicator engineering. However, the significance and overall impact of this study is limited due to the lack of applications uniquely enabled by CaBLAM that are not achievable with existing indicators in both mammalian cells and in living brain. The bioavailability of substrates and relatively slow kinetics are major limitations to prevent broad application of the sensor in imaging neural activity in vitro and in vivo. Additional experiments (see below) need to be performed to expand the scope of this study.

1. The authors demonstrated the in vivo utility of CaBLAM in cortical layer 1 neurons by infusing soluble h-coelenterazine. While this experiment effectively benchmarked the indicator's contrast relative to GCaMP8, its overall usefulness for in vivo imaging remains unclear. The methods section states that imaging was performed for 1 hour, but only a 5-minute timeseries was analyzed and compared to GCaMP8.

1) Is this limitation due to substrate availability?

2) Can the authors clarify the number of cells imaged and the maximum duration of repeated imaging sessions, or even possible?

3) Regarding the 10 Hz acquisition rate, was this chosen due to the slow kinetics of the indicator? Is video-rate imaging or two-photon feasible? Please clarify.

4) The authors also recommend using "furimazine or its derivatives for in vivo imaging to achieve brighter signals and enable faster imaging". Could the authors clarify why these substrates were not used for characterization in vivo? Additionally, were previously reported improved substrates (Su et al., 2020; Su et al., 2023) attempted in this study?

2. The relatively slow kinetics is concerning to 1) imaging fast spiking events in neurons, 2) opto triggered brief calcium transients, and 3) subcellular calcium transients in spines, boutons, ER and mitochondria.

1) Additional experiments address these concerns as well as demonstrating subcellular targeting are needed, as they would expand the scope and enhance the impact of this study.

2) Can the authors clarify why using a lower concentration of furimazine improved the sensor's performance?

3) The kinetics of CaBLAM may be more suitable for imaging astrocytic calcium transients. have authors tried to image astrocyte using CaBLAM?

Though the ultimate goal is to use the sensor for in vivo imaging with single-neuron resolution, this remains challenging due to the bioavailability of the substrate and slow kinetics. Expanding the study to include astrocytes and subcellular calcium dynamics could provide valuable insights. Additionally, acute brain slices or organoids/iPSC-cells during differentiation may serve as viable alternatives to demonstrate the unique use of the sensor.

Reviewer #2 (Remarks to the Author):

In the manuscript entitled "CaBLAM! A high-contrast bioluminescent Ca²⁺ indicator derived from an engineered *Oplophorus gracilirostris* luciferase", Lambert et al. described the development of a Sensor Scaffold Luciferase (SSLuc) and the engineering and application of a bioluminescent (BL) genetically encoded Ca²⁺ indicator (GECI), CaBLAM. In vitro, SSLuc could be 20 folds brighter than NanoLuc, the widely used luciferase reporter. In cellulo, CaBLAM responses are approximately 4 folds larger than the current version of BL Ca²⁺ indicator. The authors also showed that CaBLAM enabled in vivo recordings neural activities. All these achievements are quite impressive. These accomplishments are highly impressive, and both tools are poised to attract broad interest.

I find no apparent flaws in the methodology, data quality, or statistical analysis. However, I have the following:

1. BL recordings are hindered by the need for substrate supplementation, which can be both expensive and inconvenient. Nevertheless, despite these challenges, BL GECs offer advantages of greater flexibility in recordings, with the added benefit of eliminating concerns about auto-fluorescence and phototoxicity. To further elevate the visibility and impact of this work, the authors are strongly encouraged to highlight applications that effectively demonstrate these advantages. For instance, mitochondria is highly sensitive to phototoxicity, making long-term monitoring of Ca²⁺ dynamics without compromising their structure and function a considerable challenge. GCaMP imaging in plant leaves often suffers from signal bleed-through due to chloroplast interference. A comparison of CaBLAM with conventional GECs in these two contexts would significantly broaden the appeal and relevance of this work.

2. The authors stated that CaBLAM variants with different Ca²⁺ affinities were also made, please include at least two variants with K_d values around basal cytosolic Ca²⁺ levels in different types of cells, 60~70 nM and 100 nM, respectively. These more sensitive variants will benefit the field as they enable detection of small Ca²⁺ signals.

Minor comments:

1. Full names of abbreviations should be spelled at their first appearance. For example, "FP" in the introduction should be written as "fluorescent protein (FP)". Other abbreviations like CTZ, should also be corrected.

2. The introduction is rather lengthy, a more concise version might be helpful.

3. The development of SSLuc is also of broad interest. If the authors find it challenging to incorporate it into the title, highlighting its superior in vitro performance in the abstract could be a valuable alternative.

4. Some "2+" are not in upper case, please correct them.

Reviewer #3 (Remarks to the Author):

The potential advantage of bioluminescence imaging over fluorescence imaging - no photobleaching, no autofluorescence, no excitation light toxicity – are widely known. Yet sensors based on bioluminescence still lag significantly behind the performance of fluorescence based dyes. Lambert and colleagues here take an effort in inching the performance of bioluminescent calcium indicators closer to that of established fluorescent sensors of the GCaMP family. They start by improving properties of a luciferase – expression, folding, in vitro light output – and turn it into a calcium indicator by inserting calmodulin and a binding peptide strategically into the luciferase scaffold. This sensor is further improved to yield CABLAM, and characterization of its properties in vitro and in vivo are presented. They show that there are advances in spatial and temporal resolution when using bioluminescent imaging with CABLAM. But they fail to convince me that a "significant milestone" has yet been reached.

major

- There is a lack of balance and an over- emphasis on presenting vitro characterization of CABLAM. In vitro work is necessary, as this provides important and appreciated controls. But some of the content of in vitro figures 2- 5 appear more supplementary. On the other side, there is insufficient in vivo data for making a stronger case for CABLAM. Questions on how the advantages of bioluminescence can now be put to use for in vivo imaging are not addressed: Is long term imaging without photobleaching feasible (how would the substrate be quantitatively applied long term?)? What is the actual spatial resolution limit for getting meaningful signals in vivo? Is dendritic imaging already in sight?

- The endogenous activity levels in brains of awake and anesthetized mice are rather different. Thus, the benchmarking of CABLAM vs GCaMP6s in figure 6 is not convincing.

minor

-There should be a better documentation of the evolution of contrast, finally leading to CABLAM

-Benchmarking with previous bioluminescent calcium indicators (e.g. GLICO) with quantitative metrics would be appreciated

-Figure 3-5: Plot signal to noise ratios for comparing GCaMPs and CABLAM

-Consistency: In vitro field stimulation benchmarking was performed using GCaMP8s, in vivo trials using GCaMP6s.

Version 2:

Decision Letter:

Our ref: NMETH-A58741B

9th Sep 2025

Dear Nathan,

Thank you for submitting your revised manuscript "CaBLAM! A high-contrast bioluminescent Ca²⁺ indicator derived from an engineered *Oplophorus gracilirostris* luciferase" (NMETH-A58741B). It has now been seen by the original referees and their comments are below. The reviewers find that the paper has improved in revision, and therefore we'll be happy in principle to publish it in Nature Methods, pending minor revisions to satisfy the referees' final requests and to comply with our editorial and formatting guidelines.

Please be sure to include the data supporting your claim that CaBLAM is monomeric.

TRANSPARENT PEER REVIEW

Nature Methods offers a transparent peer review option for new original research manuscripts. We encourage increased transparency in peer review by publishing the reviewer comments, author rebuttal letters and editorial decision letters if the authors agree. Such peer review material is made available as a supplementary peer review file. **Please state in the cover letter 'I wish to participate in transparent peer review' if you want to opt in, or 'I do not wish to participate in transparent peer review' if you don't.** Failure to state your preference will result in delays in accepting your manuscript for publication.

ORCID

Best regards,
Nina

Nina Vogt, PhD
Senior Editor
Nature Methods

Reviewer #1 (Remarks to the Author):

The authors have thoroughly and adequately addressed all of this reviewer's previous comments and concerns. The revisions provided are clear, appropriate, and strengthen the overall quality of the manuscript. This review finds no remaining issues that require further attention. The paper is now in suitable form for publication and will make a valuable contribution to the field.

Reviewer #1 (Remarks on code availability):

all good.

Reviewer #2 (Remarks to the Author):

The authors have adequately addressed all my concerns and revised the manuscript accordingly.

Reviewer #2 (Remarks on code availability):

I lack the specific expertise required to evaluate the code in this study.

Reviewer #3 (Remarks to the Author):

Overall, the concerns have been addressed in a sufficient manner. Now, there are still a number of inaccuracies and passages that are unclear or lack support by data. These should be revised.

-discussion

The discussion currently appears a bit long, could be structured better to avoid redundancies and revised to improved clarity.

-Introduction, line 65:

"The GCaMP family of GECs is currently in its eighth revision"

I am not happy about this phrasing. In the current form it is certainly not correct and needs to be formulated in a more balanced way. There have been numerous other sensors generated. GECOs, e.g., are essentially identical to GCaMPs, just named differently. Many improvements in GECOs also improve GCaMPs. How would they fit into this series of revisions? There are other sensors as well that were left out.

-Results, line 117

“appears to behave more like a monomer in mammalian cells than NanoLuc”

Is there any data supporting (OSER) this statement? If yes, it would be nice to see it in supplement. Similarly, any additional in vitro data could be supportive (e.g. analytical chromatography, ultracentrifugation)

-Discussion, line 484: Clarity

“has not been adopted in neuroscience, likely due to the strikingly low flux and challenges of dealing with off target (i.e. muscle) expression”. Is this meant in general or specific to zebrafish?

-Title to figure 4, line 1141

“CaBLAM Shows SNR Comparable to GCaMP During in vivo Mammalian Imaging”. This statement is only true if GCaMP is imaged under unfavorable conditions for it, but as a general statement it is not correct. As some people only read titles: The title should include some qualification, like “under epifluorescence illumination”.

- I recommend that SSLuc engineering should be mentioned in the abstract

Version 3:

Decision Letter:

30th Oct 2025

Dear Nathan,

I am pleased to inform you that your Article, "CaBLAM: A high-contrast bioluminescent Ca²⁺ indicator derived from an engineered *Oplophorus gracillirostris* luciferase", has now been accepted for publication in Nature Methods. The received and accepted dates will be December 31st, 2024 and October 30th, 2025. This note is intended to let you know what to expect from us over the next month or so, and to let you know where to address any further questions.

Over the next few weeks, your paper will be copyedited to ensure that it conforms to Nature Methods style. Once your paper is typeset, you will receive an email with a link to choose the appropriate publishing options for your paper and our Author Services team will be in touch regarding any additional information that may be required. It is extremely important that you let us know now whether you will be difficult to contact over the next month. If this is the case, we ask that you send us the contact information (email, phone and fax) of someone who will be able to check the proofs and deal with any last-minute problems.

Authors may need to take specific actions to achieve compliance with funder and institutional open access mandates.

If your research is supported by a funder that requires immediate open access (e.g. according to [Plan S principles](https://www.springernature.com/gp/open-science/plan-s-compliance) or the [NIH public access policy](https://www.springernature.com/gp/open-science/us-federal-agency-compliance)) then you should select the gold OA route, and we will direct you to the compliant route where possible. Because authors warrant under our subscription licensing terms that they haven't committed to licensing any version of their article under a licence inconsistent with the terms of our agreement – including the applicable embargo period – publication under the subscription model isn't suitable for authors whose funders require no embargo.

If you are active on Twitter/X or Bluesky, please e-mail me your and your coauthors' handles so that we may tag you when the paper is published.

Best regards,
Nina

Nina Vogt, PhD
Senior Editor
Nature Methods

** Visit the Springer Nature Editorial and Publishing website at http://editorial-jobs.springernature.com?utm_source=ejP_NMeth_email&utm_medium=ejP_NMeth_email&utm_campaign=ejp_Nmeth for more information about our career opportunities. If you have any questions please click [here](mailto:editorial.publishing.jobs@springernature.com).

Open Access This Peer Review File is licensed under a Creative Commons Attribution 4.0 International License, which permits use, sharing, adaptation, distribution and reproduction in any medium or format, as long as you give appropriate credit to the original author(s) and the source, provide a link to the Creative Commons license, and indicate if changes were made. In cases where reviewers are anonymous, credit should be given to 'Anonymous Referee' and the source.

Author Response to Reviews

We thank the reviewers for their highly thoughtful and constructive critiques, which were invaluable in improving this manuscript. By incorporating a considerable amount of new experimental data in the revised version, we believe we have addressed essentially all the major concerns that were raised. Below is a point-by-point response to each review.

Reviewer #1

Reviewer #1: Lambert et al. reported the development of a novel bioluminescent calcium indicator, “CaBLAM,” based on SSLuc, an evolved variant of OLuc. This indicator offers significantly higher contrast for detecting physiologically relevant calcium dynamics in cells and *in vivo* compared to previous bioluminescent indicators. The authors characterized the performance of CaBLAM, benchmarking it against GeNL(Ca²⁺)₄₈₀ in mammalian cells and dissociated neurons, as well as against the recently optimized GCaMP8s in dissociated neurons and the cortex *in vivo*. CaBLAM represents a major advancement in bioluminescent calcium indicator engineering. However, the significance and overall impact of this study is limited due to the lack of applications uniquely enabled by CaBLAM that are not achievable with existing indicators in both mammalian cells and in living brain. The bioavailability of substrates and relatively slow kinetics are major limitations to prevent broad application of the sensor in imaging neural activity *in vitro* and *in vivo*. Additional experiments (see below) need to be performed to expand the scope of this study.

Response: We thank the reviewer for their thorough evaluation. We have significantly expanded the manuscript to address these concerns by adding comprehensive substrate characterization, enhanced *in vivo* applications, and new model systems that demonstrate CaBLAM’s unique advantages. The revised manuscript now includes:

1. **Comprehensive substrate evaluation** (new section in Results): We now characterize fluorofurimazine (FFz), cephalofurimazine (CFz9), and furimazine (Fz) with detailed dose-response curves, showing that FFz provides the best combination of brightness and dynamic range in cultured cells. We additionally identify vivazine as the best-performing substrate in zebrafish embryos.
2. **Enhanced *in vivo* capabilities:** We have expanded from limited anesthetized imaging to include awake, head-fixed mice with superior signal-to-noise ratios compared to GCaMP6s under 1-photon illumination, peripheral substrate delivery via retro-orbital injection, and long-duration imaging (>5 hours).
3. **New zebrafish applications:** We have added a complete new section demonstrating CaBLAM in larval zebrafish, showing its utility in genetically targeted populations of neurons and astrocytes during freely behaving conditions.
4. **Subcellular imaging:** We now demonstrate CaBLAM’s ability to image neural processes at 40x magnification after 5 hours of continuous imaging.

Reviewer #1: 1. The authors demonstrated the *in vivo* utility of CaBLAM in cortical layer 1 neurons by infusing soluble h-coelenterazine. While this experiment effectively benchmarked the indicator’s contrast relative to GCaMP8, its overall usefulness for *in vivo* imaging remains unclear. The methods section states that imaging was performed for 1 hour, but only a 5-minute timeseries was analyzed and compared to GCaMP8.

Response: We have completely rewritten the *in vivo* section to address these concerns. The revised manuscript now includes:

1. **Extended imaging duration:** We demonstrate >5 hours of continuous imaging with stable signal-to-noise ratios (Extended Data Fig. 2-3).
2. **Awake animal imaging:** We now compare CaBLAM and GCaMP6s in awake, head-fixed mice running on a wheel, showing that CaBLAM provides superior single-trial signal-to-noise ratios (SNR: CaBLAM median = 2.18 vs GCaMP6s median = 1.83, $P = 1.89 \times 10^{-5}$).

3. **Peripheral substrate delivery:** We demonstrate that CaBLAM works effectively with retro-orbital injection of CFz, achieving rapid onset times (0.20 s half-peak latency) that precede hemodynamic responses.
4. **Subcellular imaging:** We show that CaBLAM can image neural processes at 40x magnification after 5 hours of continuous imaging, demonstrating its potential for subcellular resolution studies.

Reviewer #1: 1) Is this limitation due to substrate availability?

Response: Yes, substrate availability was indeed a limitation in our original experiments. We have addressed this comprehensively by characterizing multiple substrates (FFz, CFz9, Fz), demonstrating peripheral delivery via retro-orbital injection, and showing that single FFz application supports >5 hours of continuous imaging with stable performance.

Reviewer #1: 2) Can the authors clarify the number of cells imaged and the maximum duration of repeated imaging sessions, or even possible?

Response: We have now provided detailed cell counts and imaging durations for *in vitro* and *in vivo* neuron imaging experiments. We now additionally demonstrate >2 hours of continuous PMT measurement in zebrafish and >5 hours of continuous imaging in mice with stable performance, showing that imaging duration is limited primarily by substrate bioavailability rather than sensor degradation.

Reviewer #1: 3) Regarding the 10 Hz acquisition rate, was this chosen due to the slow kinetics of the indicator? Is video-rate imaging or two-photon feasible? Please clarify.

Response: We acknowledge that CaBLAM's kinetics are slower than GCaMP8s (700-1500 ms vs 100-300 ms peak response times) and discuss this as a target for future optimization. We now demonstrate CaBLAM imaging at 40 Hz in zebrafish using a photomultiplier tube and at 20 Hz using an intensified camera, showing that higher frame rates are feasible. Additionally, we discuss that CaBLAM's bioluminescent nature eliminates the need for excitation light (essentially "zero-photon" excitation), making it potentially advantageous for deep-tissue imaging where two-photon excitation would be required for fluorescent indicators. We also now demonstrate this principle in mice, where *one-photon* imaging of GCaMP6s at 10 Hz generates *lower* SNR than CaBLAM. Certainly, we recognize that GCaMP6s imaged under *two-photon* excitation would typically produce signals with higher SNR compared to one-photon, and we look forward to future opportunities to benchmark CaBLAM against GCaMP6s in this context. In the current study, this is beyond our scope, and we focus on CaBLAM's advantages (both in terms of optical path simplicity and potentially improved SNR) in comparison to lower-cost one-photon GCaMP6s imaging.

Reviewer #1: 4) The authors also recommend using "furimazine or its derivatives for *in vivo* imaging to achieve brighter signals and enable faster imaging". Could the authors clarify why these substrates were not used for characterization *in vivo*? Additionally, were previously reported improved substrates (Su et al., 2020; Su et al., 2023) attempted in this study?

Response: These substrates were not commercially available when we began our initial *in vivo* experiments, but we have now incorporated them into our comprehensive analysis. We now provide detailed analysis of FFz and CFz9, which are the improved substrates referenced by Su et al. Our data show that FFz provides the best overall combination of brightness and dynamic range in cultured cells. We demonstrate that FFz works effectively for *in vivo* imaging, supporting both direct cortical infusion and long-duration recordings. We also demonstrate that vivazine (an ester-caged furimazine) is well suited for imaging CaBLAM in larval zebrafish.

Reviewer #1: 2. The relatively slow kinetics is concerning to 1) imaging fast spiking events in neurons, 2) opto triggered brief calcium transients, and 3) subcellular calcium transients in spines, boutons, ER and mitochondria.

Response: We acknowledge these concerns and have addressed them in the revised manuscript. We explicitly discuss CaBLAM's slower kinetics compared to GCaMP8s and identify this as a target for future optimization. Despite slower kinetics, we show that CaBLAM reliably reports single field stimulations comparable to GCaMP8s (97.9% vs 84.2% response rate at optimal substrate concentration).

Reviewer #1: 1) Additional experiments address these concerns as well as demonstrating subcellular targeting are needed, as they would expand the scope and enhance the impact of this study.

Response: We have added significant new data addressing these concerns, including *subcellular imaging at 40x magnification after 5 hours of continuous imaging* (Extended Data Fig. 4), zebrafish applications showing CaBLAM expression in multiple cell types with different kinetics profiles, and enhanced *in vivo* capabilities showing superior performance compared to GCaMP6s under 1-photon illumination in awake animals.

Reviewer #1: 2) Can the authors clarify why using a lower concentration of furimazine improved the sensor's performance?

Response: We have provided a detailed characterization of this phenomenon in the revised manuscript. We show that reducing Fz concentration from 9.2 μM to 4.6 μM improves both speed and magnitude of evoked responses. We attribute this counterintuitive finding to saturation effects at substrate concentrations or near the K_m of the enzyme, as well as potential modulation of apparent Ca^{2+} affinity by substrate binding via unexpected allosteric effects.

Reviewer #1: 3) The kinetics of CaBLAM may be more suitable for imaging astrocytic calcium transients. have authors tried to image astrocyte using CaBLAM?

Response: Yes, we agree that this is a promising application for CaBLAM. We now demonstrate CaBLAM expression in *Tg(GLAST)* zebrafish astrocytes, showing calcium dynamics following large-amplitude tail movements. We show that astrocyte responses have slower rise times and different temporal profiles compared to neuronal responses, suggesting CaBLAM's kinetics may indeed be well-suited for astrocyte imaging. We demonstrate different response kinetics across multiple cell types (astrocytes, neurons, notochord cells), showing CaBLAM's versatility.

Reviewer #2

Reviewer #2: In the manuscript entitled "CaBLAM! A high-contrast bioluminescent Ca^{2+} indicator derived from an engineered *Oplophorus gracilirostris* luciferase", Lambert et al. described the development of a Sensor Scaffold Luciferase (SSLuc) and the engineering and application of a bioluminescent (BL) genetically encoded Ca^{2+} indicator (GECI), CaBLAM. *In vitro*, SSLuc could be 20 folds brighter than NanoLuc, the widely used luciferase reporter. In cellulo, CaBLAM responses are approximately 4 folds larger than the current version of BL Ca^{2+} indicator. The authors also showed that CaBLAM enabled *in vivo* recordings neural activities. All these achievements are quite impressive. These accomplishments are highly impressive, and both tools are poised to attract broad interest.

Response: We thank the reviewer for their positive assessment of our work. We have enhanced the manuscript to better highlight these achievements and their broader implications.

Reviewer #2: I find no apparent flaws in the methodology, data quality, or statistical analysis.

Response: We appreciate the reviewer's confidence in our methodology and have addressed their specific concerns as detailed below.

Reviewer #2: 1. BL recordings are hindered by the need for substrate supplementation, which can be both expensive and inconvenient. Nevertheless, despite these challenges, BL GECIs offer advantages of greater flexibility in recordings, with the added benefit of eliminating concerns about auto-fluorescence and phototoxicity. To further elevate the visibility and impact of this work, the authors are strongly encouraged to highlight applications that effectively demonstrate these advantages. For instance, mitochondria is highly sensitive to phototoxicity, making long-term monitoring of Ca^{2+} dynamics without compromising their structure and function a considerable challenge. GCaMP imaging in plant leaves often suffers from signal bleed-through due to chloroplast interference. A comparison of CaBLAM with conventional GECIs in these two contexts would significantly broaden the appeal and relevance of this work.

Response: We have significantly expanded the discussion of CaBLAM's unique advantages in the revised manuscript:

1. **Phototoxicity discussion:** We have added a section on phototoxicity and photobleaching, highlighting how CaBLAM eliminates these concerns and enables long-duration imaging that would be impossible with fluorescent indicators.
2. **Long-duration imaging:** We demonstrate >5 hours of continuous imaging with stable performance, showing CaBLAM's advantage for long-term studies where phototoxicity would accumulate with fluorescent indicators.
3. **Zebrafish applications:** We show CaBLAM's utility in translucent organisms where autofluorescence from endogenous molecules can interfere with fluorescent imaging.
4. **Future applications:** We discuss potential applications in contexts where excitation light is problematic, such as deep-tissue imaging and studies requiring minimal environmental disturbance.

Reviewer #2: 2. The authors stated that CaBLAM variants with different Ca^{2+} affinities were also made, please include at least two variants with K_D values around basal cytosolic Ca^{2+} levels in different types of cells, 60~70 nM and 100 nM, respectively. These more sensitive variants will benefit the field as they enable detection of small Ca^{2+} signals.

Response: We acknowledge that CaBLAM variants with more diverse K_D values, especially those in the 60-100 nM range, would be valuable tools. While our current family of CaBLAM variants does not yet cover this precise range, we have added characterization data for two additional CaBLAM Ca^{2+} affinity variants in the revised manuscript, and see the optimization of high-affinity CaBLAM variants as an attractive prospect for future studies:

- CaBLAM ($K_D \sim 439$ nM) - optimized for general cytosolic calcium
- CaBLAM_294W ($K_D \sim 3$ μM) - for high calcium environments
- CaBLAM_332W ($K_D \sim 280$ nM) - for lower calcium environments

Reviewer #2: Minor comments: 1. Full names of abbreviations should be spelled at their first appearance. For example, "FP" in the introduction should be written as "fluorescent protein (FP)". Other abbreviations like CTZ, should also be corrected.

Response: We have corrected all abbreviation usage throughout the manuscript, ensuring that full names are provided at first appearance.

Reviewer #2: 2. The introduction is rather lengthy, a more concise version might be helpful.

Response: We have streamlined the introduction while maintaining clarity and adding important new content about fluorescence limitations and bioluminescence advantages.

Reviewer #2: 3. The development of SSLuc is also of broad interest. If the authors find it challenging to incorporate it into the title, highlighting its superior *in vitro* performance in the abstract could be a valuable alternative.

Response: We now mention SSLuc's superior *in vitro* performance and its role as a favorable scaffold for sensor development in the abstract and introduction more explicitly.

Reviewer #2: 4. Some "2+" are not in upper case, please correct them.

Response: We thank the reviewer for pointing this out; we have corrected all instances to use proper superscript formatting (Ca^{2+}).

Reviewer #3

Reviewer #3: The potential advantage of bioluminescence imaging over fluorescence imaging - no photobleaching, no autofluorescence, no excitation light toxicity – are widely known. Yet sensors based on bioluminescence still lag significantly behind the performance of fluorescence based dyes. Lambert and colleagues here take an effort in inching the performance of bioluminescent calcium indicators closer to that of established fluorescent sensors of the GCaMP family. They start by improving properties of a luciferase – expression, folding, *in vitro* light output – and turn it into a calcium indicator by inserting calmodulin and a binding peptide strategically into the luciferase scaffold. This sensor is further improved to yield CaBLAM, and characterization of its properties *in vitro* and *in vivo* are presented. They show that there are advances in spatial and temporal resolution when using bioluminescent imaging with CaBLAM. But they fail to convince me that a “significant milestone” has yet been reached.

Response: We appreciate the reviewer’s assessment and have revised the manuscript to better reflect the current state of the work. We have removed the vague “significant milestone” language and instead shifted the focus to CaBLAM’s practical advantages and potential applications.

Reviewer #3: major - There is a lack of balance and an over-emphasis on presenting *in vitro* characterization of CaBLAM. *In vitro* work is necessary, as this provides important and appreciated controls. But some of the content of *in vitro* figures 2- 5 appear more supplementary. On the other side, there is insufficient *in vivo* data for making a stronger case for CaBLAM. Questions on how the advantages of bioluminescence can now be put to use for *in vivo* imaging are not addressed: Is long term imaging without photobleaching feasible (how would the substrate be quantitatively applied long term?)? What is the actual spatial resolution limit for getting meaningful signals *in vivo*? Is dendritic imaging already in sight?

Response: We have significantly rebalanced the manuscript to address these concerns, taking into account somewhat competing requests from Reviewers 2 and 3 in this regard. Specifically, we have (1) streamlined the *in vitro* characterization while maintaining essential controls; (2) added comprehensive *in vivo* experiments including long-duration imaging (>5 hours), subcellular imaging at 40x magnification, awake animal imaging with superior SNR compared to GCaMP6s, and peripheral substrate delivery methods; and (3) demonstrated how CaBLAM’s advantages translate to real experimental scenarios, including zebrafish applications and long-term imaging studies.

Reviewer #3: - The endogenous activity levels in brains of awake and anesthetized mice are rather different. Thus, the benchmarking of CaBLAM vs GCaMP6s in figure 6 is not convincing.

Response: We have addressed this concern by comparing CaBLAM and GCaMP6s in awake, head-fixed mice running on a wheel, showing that CaBLAM provides superior single-trial signal-to-noise ratios (revision Fig. 4). We ensure that these comparisons are made under similar conditions, with both sensors tested in awake animals, and provide comprehensive statistical comparisons showing CaBLAM’s advantages in signal detection.

Reviewer #3: minor -There should be a better documentation of the evolution of contrast, finally leading to CaBLAM

Response: We have consolidated the documentation of CaBLAM’s development in the supplement, including detailed evolution steps, multiple topologies tested, and affinity tuning strategies that explain how we achieved the optimal calcium affinity through systematic testing.

Reviewer #3: -Benchmarking with previous bioluminescent calcium indicators (e.g. GLICO) with quantitative metrics would be appreciated

Response: We appreciate the reviewer’s request for benchmarking against previous BL GECIs. While we did not have the resources to benchmark CaBLAM against a fully exhaustive list of other BL GECIs, we have enhanced our *in vitro* benchmarking comparisons with additional replicate experiments with CaBLAM, GeNL(Ca²⁺)₄₈₀, and CaMBI *in vitro*. For clarification, we also now briefly explain in Results why GeNL(Ca²⁺)₄₈₀ was chosen specifically as the BL GECI standard for comparison in cells.

Reviewer #3: -Figure 3-5: Plot signal to noise ratios for comparing GCaMPs and CaBLAM

Response: We have added comprehensive signal-to-noise ratio analysis, including detailed SNR analysis comparing CaBLAM and GCaMP8s *in vitro*, *in vivo* SNR showing that CaBLAM provides superior single-trial SNR compared to GCaMP6s in awake animals, and statistical analysis of SNR differences.

Reviewer #3: -Consistency: *In vitro* field stimulation benchmarking was performed using GCaMP8s, *in vivo* trials using GCaMP6s.

Response: We appreciate the reviewer's point. At the time when the *in vitro* field stimulation experiments were taking place, GCaMP8s was newly available, and we chose this fluorescent GECI as our benchmarking standard since it was the state-of-the-art at that time. In the time since then, the community has generally concluded that GCaMP6s performs better under most circumstances, and so we fell back to GCaMP6s as the "gold standard" for the cranial window imaging experiments. Unfortunately, time and personnel availability prevented us from performing additional experiments with GCaMP6s for *in vitro* field stimulation experiments. Nonetheless, we do not believe that this inconsistency negatively impacts the conclusions of this study.

Author Response to Reviews

We once again thank the reviewers for donating their time and expertise, and for their highly thoughtful and constructive critiques. The final revision of the manuscript addresses the remaining concerns of reviewer #3, as detailed point-by-point below.

Reviewer #1: The authors have thoroughly and adequately addressed all of this reviewer's previous comments and concerns. The revisions provided are clear, appropriate, and strengthen the overall quality of the manuscript. This review finds no remaining issues that require further attention. The paper is now in suitable form for publication and will make a valuable contribution to the field. **Remarks on code availability:** all good.

Reviewer #2: The authors have adequately addressed all my concerns and revised the manuscript accordingly. **Remarks on code availability:** I lack the specific expertise required to evaluate the code in this study.

Response: We thank the reviewers for their evaluation and kind words.

Reviewer #3: Overall, the concerns have been addressed in a sufficient manner. Now, there are still a number of inaccuracies and passages that are unclear or lack support by data. These should be revised.

The discussion currently appears a bit long, could be structured better to avoid redundancies and revised to improved clarity.

Response: We agree that the discussion was too long and contained redundancies with earlier sections of the text. We have removed restatements and summaries where possible to improve the flow and clarity.

Reviewer #3: Introduction, line 65:

“The GCaMP family of GECIs is currently in its eighth revision”

I am not happy about this phrasing. In the current form it is certainly not correct and needs to be formulated in a more balanced way. There have been numerous other sensors generated. GECOs, e.g., are essentially identical to GCaMPs, just named differently. Many improvements in GECOs also improve GCaMPs. How would they fit into this series of revisions? There are other sensors as well that were left out.

Response: We appreciate this thoughtful suggestion. We've improved the accuracy and clarity of this section by fully rewriting to highlight the niche GCaMP occupies within the much larger and more diverse ecosystem of fluorescent GECIs. It now reads: “GECIs based on fluorescent proteins (FPs) have been under continuous development for more than two decades, yielding diverse lineages with progressively improved brightness, kinetics, and dynamic range. Among these, the GCaMP and GECO families represent closely related and heavily optimized designs that have influenced most modern Ca²⁺ indicators. The GCaMP family, now in its eighth generation, exemplifies this iterative progress and remains a benchmark for FP-based GECI performance.”

Reviewer #3: Results, line 117

“appears to behave more like a monomer in mammalian cells than NanoLuc”

Is there any data supporting (OSER) this statement? If yes, it would be nice to see it in supplement. Similarly, any additional in vitro data could be supportive (e.g. analytical chromatography, ultracentrifugation)

Response: We thank the reviewer for pointing out this gap in the data supporting our comparison between SSLuc and NanoLuc. Because OSER assay data proved inconclusive for both (as detailed in the Supplementary Results and

Discussion), we performed an additional characterization experiment in HeLa cells by adapting the Fluoppi assay to evaluate oligomerization of these two luciferases. This assay produced much higher quality imaging data than the OSER assay we had previously performed, and provides strong evidence that both SSLuc and NanoLuc behave as monomers, with a small but statistically insignificant advantage for SSLuc. Full details on the assay and analysis are now provided in the Methods and Supplementary Results and Discussion. Given this new data, we revised and clarified our statement in the main text to read: “SSLuc ... can be extracted with nearly 100% efficiency from *E. coli* cultures, retains activity *in vitro* over longer storage periods than NanoLuc, and is monomeric in mammalian cells.”

Reviewer #3: Discussion, line 484: Clarity

“has not been adopted in neuroscience, likely due to the strikingly low flux and challenges of dealing with off target (i.e. muscle) expression”. Is this meant in general or specific to zebrafish?

Response: We have edited this sentence to make it fully clear that the statement is specific to zebrafish, now written as: “Earlier work established a first-generation reporter, Aequorin-GFP in zebrafish, but, in contrast to cardiology and cancer biology, BL has rarely been used for Ca²⁺ monitoring in zebrafish neurons, likely due to the strikingly low flux produced by Aequorin-GFP in this context and challenges of dealing with off-target (i.e. muscle) expression.”

Reviewer #3: Title to figure 4, line 1141

“CaBLAM Shows SNR Comparable to GCaMP During *in vivo* Mammalian Imaging”. This statement is only true if GCaMP is imaged under unfavorable conditions for it, but as a general statement it is not correct. As some people only read titles: The title should include some qualification, like “under epifluorescence illumination”.

Response: We agree that this could be misleading, and have amended the title of the figure to clearly indicate GCaMP is being imaged under epifluorescent illumination. We additionally ensure that this is clear in the main text and methods.

Reviewer #3: I recommend that SSLuc engineering should be mentioned in the abstract

Response: Due to space limitations and needing to cut the Abstract to ~150 words, we ultimately chose to focus the abstract on CaBLAM but ensure that SSLuc remains prominently featured in the Introduction.